# Ultrasensitive mechanical/thermal response of a P(VDF-TrFE) sensor with a tailored network interconnection interface

Bo Li ®[1] ✉, Chuanyang Cai[2], Yang Liu ®[3], Fang Wang[4], Bin Yang[1], Qikai Li[1], Pengxiang Zhang[1], Biao Deng[1], Pengfei Hou ®[2] ✉ & Weishu Liu ®[1] ✉

Ferroelectric polymers have great potential applications in mechanical/thermal sensing, but their sensitivity and detection limit are still not outstanding. We propose interface engineering to improve the charge collection in a ferroelectric poly(vinylidene fluoride-co-trifluoroethylene) copolymer (P(VDF-TrFE)) thin film via cross-linking with poly(3,4-ethylenedioxythiophene) doped with polystyrenesulfonate (PEDOT:PSS) layer. The as-fabricated P(VDF-TrFE)/PEDOT:PSS composite film exhibits an ultrasensitive and linear mechanical/thermal response, showing sensitivities of 2.2 V kPa$^{-1}$ in the pressure range of 0.025–100 kPa and 6.4 V K$^{-1}$ in the temperature change range of 0.05–10 K. A corresponding piezoelectric coefficient of −86 pC N$^{-1}$ and a pyroelectric coefficient of 95 μC m$^{-2}$ K$^{-1}$ are achieved because more charge is collected by the network interconnection interface between PEDOT:PSS and P(VDF-TrFE), related to the increase in the dielectric properties. Our work shines a light on a device-level technique route for boosting the sensitivity of ferroelectric polymer sensors through electrode interface engineering.

Ferroelectric polymers have attracted enormous attention for their promising applications in electromechanical and electrothermal correction, such as in sensors[1], actuators[2], and energy harvesting[3], because of their chemical stability, flexibility, low-temperature processability, biocompatibility, etc[4–7]. The most widely used ferroelectric polymer-based sensors contain poly(vinylidene fluoride) (PVDF) or its copolymers, which translate pressure, temperature, strain, and vibration stimuli into free surface-bound charge based on pyroelectric[8], piezoelectric[9], and capacitive[10] effects, and a pair of top and bottom planar electrodes transforms this charge into an external load[11–13]. This indicates that the sensing properties of ferroelectric polymer devices depend on the piezoelectric coefficient ($d_{33}$), pyroelectric coefficient ($p$), and interface between electrodes and polymers. The conventional poly(vinylidene fluoride-co-trifluoroethylene) copolymer (P(VDF-

TrFE)) has a low sensor voltage output because of its relatively low piezoelectric and pyroelectric coefficients, resulting in the sensor not meeting the demand of a high voltage output for high sensitivity sensing. Therefore, the piezoelectric and pyroelectric coefficients of P(VDF-TrFE) are critical to improving the sensitivity and detection limit of P(VDF-TrFE)-based sensors.

Boosting the piezoelectric and pyroelectric coefficients of polymers is widely applied to enhance the sensing characteristics of P(VDF-TrFE)-based sensors, including incorporating inorganic fillers with high piezoelectric or pyroelectric coefficients with the polymer[14,15] increasing the degree of β-phase crystallinity using mechanical stretching[16,17], applying electrostatic spinning[18–20], etc. For example, a $d_{33}$ of −63.5 pC N$^{-1}$ was reported by using the morphotropic phase boundary effect of the P(VDF-TrFE) ($C_{VDF} = 50$ mol%) copolymer[21].

[1]Department of Materials Science and Engineering, Southern University of Science and Technology, Shenzhen, Guangdong 518055, China. [2]School of Materials Science and Engineering, Xiangtan University, Hunan Xiangtan 411105, China. [3]State Key Laboratory of Material Processing and Die & Mould Technology, School of Materials Science and Engineering, Huazhong University of Science & Technology, Wuhan 430074, China. [4]Institute of Biomedical & Health Engineering, Shenzhen Institute of Advanced Technology (SIAT), Chinese Academy of Sciences (CAS), Shenzhen 518055, China. ✉e-mail: lib6@sustech.edu.cn; houpf@xtu.edu.cn; liuws@sustech.edu.cn

A highly piezoelectric polymer with a $d_{33}$ of −62 pC N$^{-1}$ was discovered based on a poled biaxially oriented PVDF sheet with pristine crystals[22]. However, there is a lack of effort to go beyond these classic strategies to increase the sensitivity of the mechanical/thermal response. In contrast, the interface between ferroelectric polymers and their widely used metal electrodes, as in a voltage-response sensor device, is far from being well understood. The low surface energy of fluoropolymers makes these polymers not wettable with any kind of liquid and not able to stick to any other solid materials, especially metals, which are widely used as electrodes. This results in regular air gaps between the polymers and metal electrodes, which lead to degradation of the pyroelectric and piezoelectric output voltages[23,24]. As a result, the piezoelectric effect is frequently hampered by poor adhesion with metal electrodes causing higher interfacial impedance with the electrodes and a lack of flexibility to appreciably bend[25]. An enhancement in the adhesion between electrodes and nanofibers was observed by penetrating the metal electrodes into BaTiO$_3$/P(VDF−TrFE) composite nanofibers due to the enlargement of the contact area between the polymers and electrodes[26,27]. Furthermore, the heat diffusion associated with the radiation absorption coefficient of the metal electrodes decreases the magnitude of the pyroelectric voltage[28–30]. Improving the electrode interface for mechanical/thermal sensors is critical. The sensitivity has been reported to be significantly increased through engineering of the electrode interface in capacitance-response sensors by introducing electron double layers, an intrafillable architecture, electrostatic interactions, etc[31–33]. However, in voltage-response sensors, the effect of the electrode interface on the sensitivity under mechanical/thermal stimulation is still an open question.

Here, we report an electrode interface engineering strategy to significantly increase the sensitivity of ferroelectric polymer P(VDF-TrFE)-based mechanical/thermal sensors by using the conductive polymer PEDOT:PSS (PEDOT, poly(3,4-ethylenedioxythiophene); PSS, polystyrene sulfonic acid) as the electrode. We claim that the tailored three-dimensional electrode interface is characterized by a formed network interconnection interface (NII) structure, increasing the charge collection and hence improving the voltage response under mechanical/thermal stimulation. Phase field simulation confirmed that the tailored NII structure increases the pyroelectric and piezoelectric coefficients. The as-fabricated P(VDF-TrFE)/PEDOT:PSS composite film exhibits an ultrasensitive and linear mechanical/thermal response, showing sensitivities of 2.2 V kPa$^{-1}$ in the pressure range of 0.025–100 kPa and 6.4 V K$^{-1}$ in the temperature change range of 0.05–10 K. The corresponding piezoelectric coefficient is −86 pC N$^{-1}$, and the pyroelectric coefficient is 95 μC m$^{-2}$ K$^{-1}$, which are outstanding among the reported P(VDF-TrFE)-based mechanical/thermal sensors. Our work shines a light on electrode interface engineering to increase the sensitivity of voltage-response sensors, providing an alternative to structure tailoring of ferroelectric polymers.

## Results

A schematic diagram of the P(VDF-TrFE)/PEDOT:PSS composite film and the corresponding test system can be found in Fig. S1. The NII structure between P(VDF-TrFE) and PEDOT:PSS was tailored with dimethyl sulfoxide (DMSO) solvent through interdiffusion, as schematically depicted in Fig. 1a–c. Dissolution of PEDOT:PSS in DMSO leads to the formation of PEDOT:PSS clusters (Fig. S2a). The NII structure in the composite film is formed by the interweaving of these PEDOT:PSS clusters with P(VDF-TrFE) chains. The formation of an NII structure do not changes the typical three-layer thin film structure. Figure 1d, e (left) shows cross-sectional scanning electron microscopy (SEM) images of the composite film with PEDOT:PSS electrode (~3.3 μm), and the fluorine distribution obtained by energy dispersive spectroscopy is shown in Fig. 1e (right). The NII structure with an average thickness of 8 μm is identified between the P(VDF-TrFE) and PEDOT:PSS layers. Then, sulfur trioxide (SO$_3$) was chosen as the marker

of PEDOT:PSS to examine the formation of the NII layer by time-of-flight secondary ion mass spectrometry (TOF-SIMS). The continuous SO$_3$ signal (Fig. 1f) detected in P(VDF-TrFE) confirms that PEDOT:PSS penetrated into P(VDF-TrFE). It has been proved that the NII thickness can be obtained for the brightness of the cross-section of the composite film where the NII is darker than the P(VDF-TrFE). X-ray diffraction of P(VDF-TrFE) and P(VDF-TrFE)/PEDOT:PSS shows that the strongest diffraction at $2\theta = 20.5$ °C from $\beta$-phase with (110)/(200) planes (Fig. S2b). The polarization-electric field hysteresis loops of the P(VDF-TrFE) films with Au ($|d_{33}| = 20$ pC N$^{-1}$), PEDOT:PSS electrode proved that the composite film has ferroelectric properties (Fig. S3). The leakage current of the composite film increases with the increase of $d_{33}$, and the reason will be explained later. Figure 1g shows the piezoelectric coefficient ($d_{33}$) of the P(VDF-TrFE) film we prepared with PEDOT:PSS and different metal electrodes (Table S1). Without electrode indicates that no electrodes were used for $d_{33}$ measurement to provide a reference to show the effect of different electrodes on $d_{33}$ values. The maximum absolute value of $d_{33}$ measured for the P(VDF-TrFE)/PEDOT:PSS composite film is 86 pC N$^{-1}$, which is over four times larger than the value obtained for the intrinsic P(VDF-TrFE) film ($|d_{33}|$ -20 pC N$^{-1}$) and other works (Fig. 1h)[34–38]. In the P(VDF-TrFE)/PEDOT:PSS composite film with NII, the charges generated by pressure and temperature are trapped inside the NII in addition to those accumulated on the surface of the composite film. The electromechanical and electrothermal energy conversion is enhanced when more charges accumulate (Fig. S4a). In contrast, charges in conventional P(VDF-TrFE) devices with metal electrodes accumulate only on the surfaces (Fig. S4b). Meanwhile, the network interconnection between PEDOT:PSS and P(VDF-TrFE) results in an excellent mechanical coupling effect that enables a broader response to a larger variety of pressure-induced strains. The increased contact area between PEDOT:PSS and P(VDF-TrFE) also reduces the total resistance of the composite film[39], resulting in improved piezoelectric properties. The pyroelectric coefficient obtained from the pyroelectric current (see supplementary information for calculation details and the data listed in Table S2) is 95 μC m$^{-2}$ K$^{-1}$, which is much higher than that of intrinsic P(VDF-TrFE) (27 μC m$^{-2}$ K$^{-1}$) and other pyroelectric polymers due to the NII enhances charge collection, as shown in Fig. 1i[40–44]. The PEDOT:PSS electrode allows the composite film to absorb more radiant heat energy compared to the Au electrode due to the higher absorption performance[45,46]. Subsequently, the pyroelectric response was drastically improved by increasing the derivative of temperature with respect to time ($dT/dt$).

A phase field simulation was carried out to illustrate the mechanism of the NII regarding the piezoelectric and pyroelectric properties of the P(VDF-TrFE)/PEDOT:PSS composite film. Fig. S5 shows the simulated polarization distribution of the polarized P(VDF-TrFE) films without and with NII, respectively, where the color indicates the magnitude and direction of polarization. The calculated piezoelectric coefficient and pyroelectric properties increase with increasing NII thickness ratio, which is in accord with our experimental results (Fig. 2a). In the calculation, the piezoelectric coefficient is determined by the permittivity[47] (see the simulation section). The calculated permittivity shows a monotonically increasing tendency as a function of NII thickness (Fig. S6). According to the Koop's model[48], the dielectric heterogeneous network conductivity can be considered as a conducting PEDOT:PSS separated by the high-resistance P(VDF-TrFE). The space charge created by the stress due to the piezoelectric effect builds up at the P(VDF-TrFE), controlling the available free charge carriers at the P(VDF-TrFE) and leading to a higher dielectric constant. Then, the capacitance of the composite film was measured (Fig. 2c). The capacitance of the composite film is higher than that of P(VDF-TrFE) with metal electrodes and increases with the thickness of the PEDOT:PSS layer. As the capacitance of a device is given as $C = \varepsilon_r \varepsilon_0 S / d$, where $\varepsilon_r$ is the relative permittivity, $\varepsilon_0$ is the vacuum permittivity, $S$ is

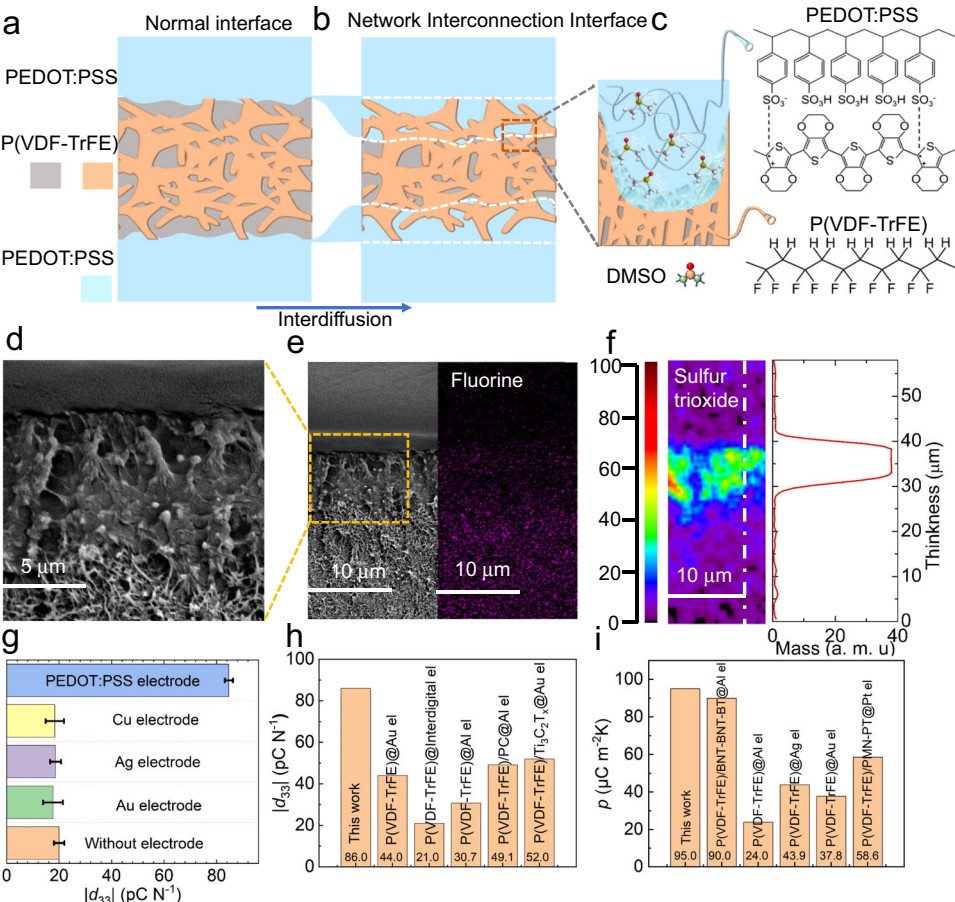

**Fig. 1 | Formation of the NII structure in a P(VDF-TrFE)/PEDOT:PSS composite film and its piezoelectric and pyroelectric properties.** Schematic illustration of the formation of the network interconnection interface in the P(VDF-TrFE)/PEDOT:PSS composite film. **a** P(VDF-TrFE)/PEDOT:PSS composite film normal interface. **b** DMSO promotes the interdiffusion process. **c** Schematic diagram of the formed NII structure. **d** and **e** SEM images of the cross-section of the P(VDF-TrFE)/PEDOT:PSS composite film and energy dispersive spectrum of fluorine. **f** Two-dimensional distribution of $SO_3$ in the interfacial components obtained by TOF-SIMS. **g** Piezoelectric coefficient of P(VDF-TrFE) with different electrode materials. The error bars represent the standard deviation estimated from at least three measurements with different samples. Summary of the **h** piezoelectric coefficient and **i** pyroelectric coefficient. Polycarbonate (PC), $(Bi_{0.5}Na_{0.5})TiO_3$-$(Bi_{0.5}K_{0.5})TiO_3$-$BaTiO_3$ (BNT-BKT-BT), 0.65PMN-0.35PT (PMN-PT), electrode (el).

the contact area, and $d$ is the distance between the top and bottom electrodes. An increase in the capacitance with the P(VDF-TrFE) confirms an increase in the permittivity. In addition, the effective contact area between the P(VDF-TrFE) and PEDOT:PSS increase due to the heterogeneous network conductivity. This suggests that the composite film has a form of network interconnection conductivity that increases permittivity[26,49,50] by increasing the contact area between the polymer and the electrodes. The NII formed during DMSO dissolution improves the capacitance of the composite film, as it enhances the direct mechanical interlocking between PEDOT:PSS and P(VDF-TrFE) and enlarges the contact area between the two materials. Then, the effect of PEDOT:PSS solution on NII thickness was investigated. Fig. S7 shows the NII thickness as a function of PEDOT:PSS thickness with PEDOT:PSS volume fraction of 5 vol% which is obtained from a SEM image of the cross-section of the P(VDF-TrFE)/PEDOT:PSS composite film. The NII thickness increases when the PEDOT:PSS thickness increases, which is controlled by the volume of the applied PEDOT:PSS solution and the DMSO volume fraction. Figure 2d demonstrates that the $|d_{33}|$ of the composite film increases when the NII electrode thickness increases, which is caused by the increase in the NII thickness. Figure 2e shows $|d_{33}|$ as a function of the DMSO volume fraction with different composite film thickness (Table S3). The $|d_{33}|$ of the composite film depends on the DMSO volume fraction, in which $|d_{33}|$ fluctuates with increasing DMSO volume and reaches a maximum value of 86 pC $N^{-1}$ when the DMSO volume fraction is 20 vol % with the P(VDF-TrFE)/PEDOT:PSS thickness of 230 μm. When the DMSO volume fraction is 25 vol% for various composite film thicknesses, $|d_{33}|$ significantly declines with a further increase in the composite film thickness. At a high DMSO volume fraction, the NII may contact and lead to neutralization of the positive and negative charges (Fig. S8a), eventually leading to a decrease in $d_{33}$ and an increase in leakage current. This was confirmed by the calculated dimensionless surface potential obtained from Maxwell's equation (Fig. S8b), which decreases when the NII thickness ratio increases from 81.3% to 87.5%. Other solutions that can dissolve both P(VDF-TrFE) and PEDOT:PSS are suspected to also possibly form the NII and boost the piezoelectric properties. Dimethylformamide (DMF) was then applied as the solution during the preparation of the PEDOT:PSS electrode. Figure 2f shows the $|d_{33}|$ of the composite film as a function of the electrode thickness with DMF (5 vol%) and DMSO (5 vol%). DMF can enhance the $|d_{33}|$ of the composite film, but not compared with DMSO. Because DMSO can enhance the conductivity of PEDOT:PSS[51,52], more charge can be collected. In addition, the high boiling point of DMSO (190 °C) leads to higher corrosion of P(VDF-TrFE) with greater NII thickness compared to DMF (155 °C), as shown in Fig. S9.

The output signal of the P(VDF-TrFE)/PEDOT:PSS film as a sensor was studied with a total thickness of 230 μm and $|d_{33}|$ = 86 pC $N^{-1}$, and the solvent of PEDOT:PSS was DMSO. Figure 3a shows the signals

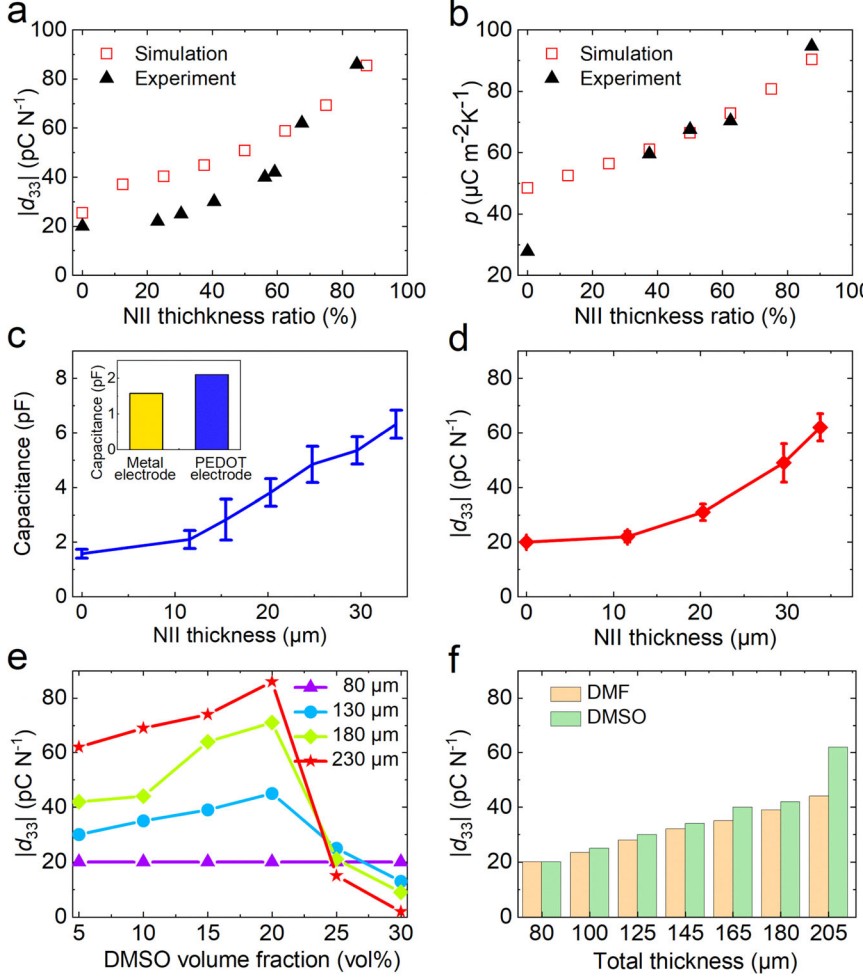

**Fig. 2 | Mechanisms of the NII in performance improvement.** Calculated and experimentally measured **a** piezoelectric coefficient and **b** pyroelectric coefficient of composite films at different NII thickness ratios. **c** Variation in the capacitance of the composite P(VDF-TrFE)/PEDOT:PSS film versus the NII thickness. The error bars represent the standard deviation estimated from at least three measurements with different samples. Inset: Comparison of the capacitance of a P(VDF-TrFE) film with metal and PEDOT:PSS electrodes. **d** Piezoelectric coefficient of the composite film versus the thickness of the NII with the DMSO volume fraction of 5 vol%. The error bars represent the standard deviation estimated from at least three measurements with different samples. **e** Piezoelectric coefficient of the composite film as a function of the DMSO volume fraction. **f** Comparison of piezoelectric coefficient of the composite film prepared using DMSO solution (5 vol%) and DMF solution (5 vol%).

output by the composite film (top) and P(VDF-TrFE) with Au electrodes (bottom) in response to force loading and unloading at 0.75 Hz for a pressure of 100 kPa, which indicates that the pressure response range and voltage of the composite film are better than those of P(VDF-TrFE) with metal electrodes. The rise time of the composite film, defined as the time required for the response voltage to rise from 10% to 90%[53] (0.06 S) (Fig. S10a and S10b), is shorter than that of the metal electrode (0.1 S), which is partly due to the higher rate of domain evolution in the composite film (Fig. S11, see supplementary information for details). This indicates that the composite film has a better piezoelectric response. Figure 3b partially shows that the amplitude of the piezoelectric output signals increases in step with the applied pressure in the 0.025–100 kPa range and the current was shown in Fig. S12a. In Fig. 3c, the output voltage of the composite film is plotted as a function of applied pressure, and the piezoelectric response at low pressure of 25–100 Pa is shown in Fig. S13. These results confirm that the composite film exhibits a linear relationship between these two variables. This linearity makes processing and calibration of the sensing signal easier, which enables more accurate pressure estimation. The pressure sensitivity ($S_p$), defined as the slope of the graph in Fig. 3c (see supplementary information for calculation details and the data listed in Table S4), is 2.2 V kPa$^{-1}$, and the coefficient of determination ($R^2_{pp}$, see the supplementary information) for the pressure sensitivity was

determined to be 0.9989. The pressure response range and peak voltage of the composite film are better than those of other P(VDF-TrFE) with PEDOT:PSS electrode[54]. The output voltage of the composite film under the studied temperatures is larger than that of P(VDF-TrFE) with metal electrodes to which a laser is applied as the heating source (Fig. 3d). The response time of output signal of the composite film induced by thermal stimulus (1.77 S) is shorter than that of the Au electrode (3.80 S) (Fig. S10c and S10d) because the PEDOT:PSS electrode can absorb more radiant heat energy and increase the d$T$/d$t$. The composite exhibits a linear response in the 0.05–10 K temperature change range, with sensitivity $S_{tp}$ = 6.4 V K$^{-1}$ (see supplementary information for details), as shown in Fig. 3e, f (Table S4). The pyroelectric current was shown in Fig. S12b. This pressure and temperature sensitivity is even better than that of the doped P(VDF-TrFE) film reported in the literature[33–36,55–60] (Fig. 3g, h), indicating that the composite film has good energy conversion properties. Note that the piezoelectric/pyroelectric current depends in part on the effective surface area of the sensor. The comparison of piezoelectric and pyroelectric current density (Fig. S14) indicates that the composite film has high piezoelectric and pyroelectric output properties. Figure 3i shows the output signal of the P(VDF-TrFE)/PEDOT:PSS composite film during a long-term durability test conducted at a pressure of 100 kPa. The response curve maintains its shape over 1000 loading cycles, and the

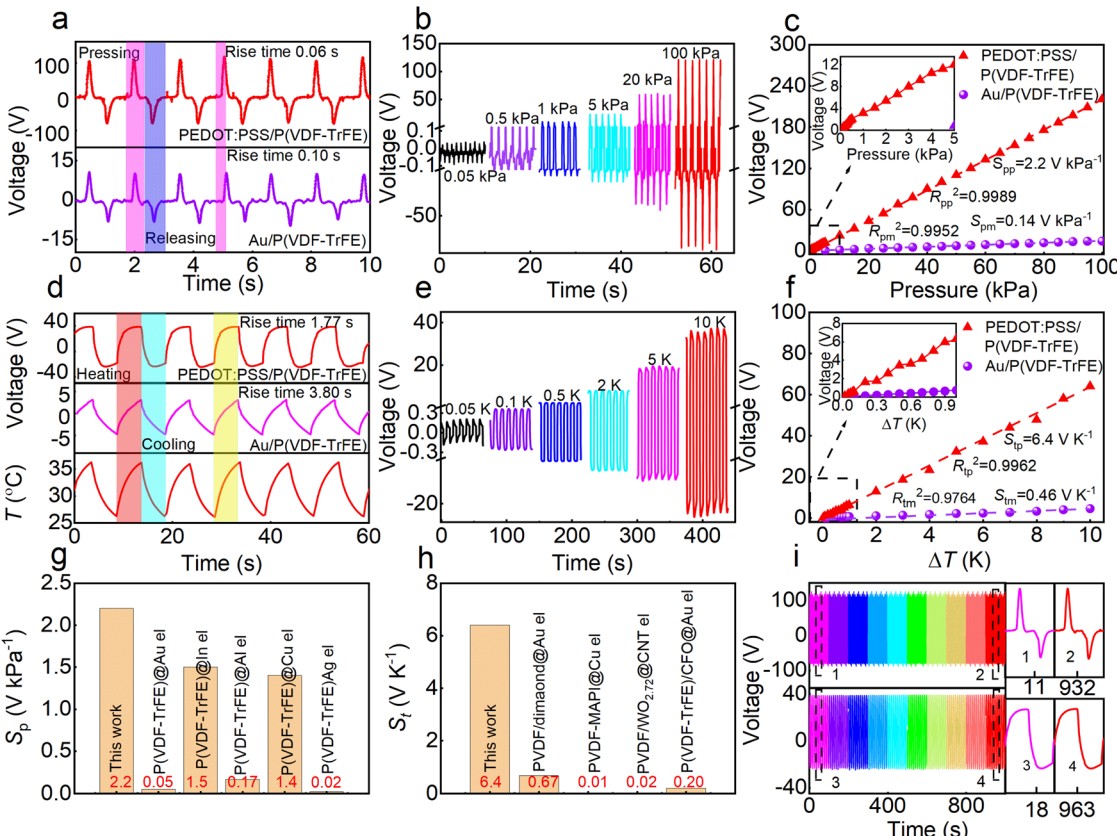

**Fig. 3 | Piezoelectric and pyroelectric voltage output of PEDOT:PSS/P(VDF-TrFE) with the NII. a** Response of the composite film (top) and P(VDF-TrFE) with Au electrode (bottom) to periodic pressing and release at 100 kPa. **b** Response of the composite film to dynamic loading with pressures in the range of 0.025–100 kPa. **c** Variation in the sensor output voltage with pressure. **d** Response of composite film (top) and P(VDF-TrFE) with Au electrode (bottom) to periodic heating and cooling at 10 K. **e** Response of the composite film to periodic heating with temperature increases in the 0.05–10 K range. **f** Variation in the sensor output voltage with temperature. Summary of **g** piezoelectric and **h** pyroelectric sensitivity. **i** The long-term stability of the piezoelectric and pyroelectric responses of the composite film in response to an applied pressure of 100 kPa (top) and the pyroelectric response to a temperature of 10 K (bottom). $CH_3NH_3PbI_3$ (MAPI), aligned carbon nanotube (CNT), $CoFe_2O_4$ (CFO), interdigital (In).

output voltage is maintained in the ~220 V range (top of Fig. 3i), indicating that the piezoelectric response of the device is reliable. The pyroelectric output voltage remains constant at ~62 V after more than 1000 cycles of heating by 10 K (down of Fig. 3i). Then, the mechanical reliability of PEDOT:PSS as a flexible electrode was tested, as shown in Fig. S15. The resistivity of the flexible electrode increases from 68 Ω cm⁻¹ to 85.5 Ω cm⁻¹ after 10000 bending cycles (Fig. S15a and S15b), indicating that the PEDOT:PSS electrode has outstanding reliability. Atomic force microscopy of the composite film confirms that the PEDOT:PSS electrode has a smooth and uniform surface with no groove defects before (Fig. S15c) and after (Fig. S15d) bending. The pressure and temperature sensor performance of the devices with and without encapsulation shows that PDMS slightly reduces the output voltages for both pyroelectric and piezoelectric devices (Fig. S16).

Next, the fabricated real-time physiological monitoring device was placed on an adult human to demonstrate the potential and usability of the composite film in health monitoring. The device was attached to the thumbs up or wrist of various volunteers to detect subtle fluctuations in arterial blood pressure (Fig. 4a). Frequency and amplitude of pulses which obtained form are accurately reproduced in terms of distance between adjacent peaks and average peak amplitude in real time. Normal heart rate remains at 72 and 79 beats per minute (bpm) for different volunteers. The measured values are close to the performance of a commercially available device (inset in Fig. 4a), demonstrating the ability of the prepared device based on a composite film for real-time physiological monitoring. The detailed waveforms in Fig. 4b show characteristic peaks corresponding to percussion waves

(p-waves), tidal waves (t-waves), and dicrotic waves (d-waves) in the human pulse, consistent with other work[61,62]. The prepared sensor was attached to the participant's throat to test the possibility of devices as a word recognition system. Vibration of the throat was detected when different words were spoken. A single peak of output voltage was present when the monosyllabic word "Hi" was spoken, whereas the multisyllabic word "Hello" elicited a multimodal shape (Fig. 4c). Finally, the sensor was used to monitor temperature and stress (Fig. S17), finger flexion (Fig. S18a) and hand grasping (Fig. S18b and S18c). In addition, a sensor array was prepared to prove that the sensor array can map the dispersion of tactile sensations (Figs. S19, S20).

In summary, a composite film with high piezoelectric coefficient (−86 pC N⁻¹) and pyroelectric coefficient (95 μC m⁻² K⁻¹) was prepared using a PEDOT:PSS penetrating electrode. A network-like interconnection interface between P(VDF-TrFE) and PEDOT:PSS was formed in the composite film. The network interconnection interface improves the charge collection of the composite film and the dielectric properties, which increases the piezoelectric and pyroelectric coefficient and is confirmed by the phase-filed simulation. In addition, the P(VDF-TrFE)/PEDOT:PSS composite film exhibits high pressure sensitivity (2.2 V kPa⁻¹) and high temperature sensitivity (6.4 V K⁻¹). It has been demonstrated that the P(VDF-TrFE)/PEDOT:PSS composite film can be used to detect physical motion and physiological signals. This work investigates the underlying mechanism of the penetrating electrode in ferroelectric copolymers and has shown that it is a promising candidate for applications in health monitoring and human-computer interaction.

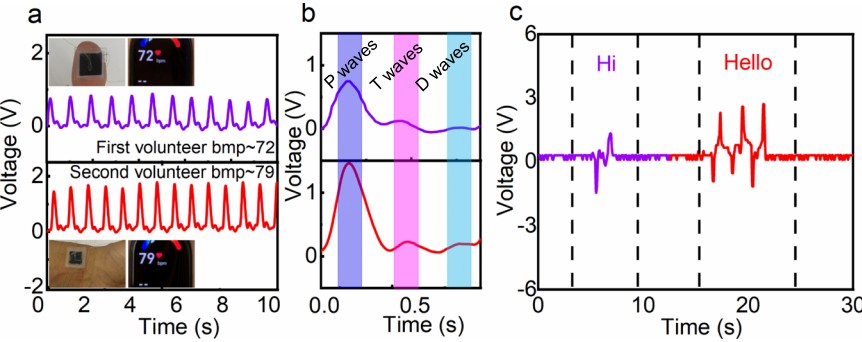

**Fig. 4 | Application of the P(VDF-TrFE)/PEDOT:PSS composite film as an ultrasensitive sensor. a** Pulse waveform from sensor monitoring of a thumb (top) and subject's wrist (bottom). **b** Magnified view of one cycle of the electrical signal. **c** Speech recognition based on monitoring laryngeal activity while speaking different words.

## Methods

### Materials

P(VDF-TrFE) was purchased from Arkema Chemistry, and DMSO and DMF were supplied by Shanghai Aladdin Biochemical Co., Ltd. PEDOT:PSS was purchased from Xi'an Baolite Photoelectric Technology Co., Ltd. Polyvinylpyrrolidone (PVP) was obtained from Shanghai Macklin Biochemical Co., Ltd.

### P(VDF-TrFE) film synthesis

To fabricate a P(VDF-TrFE) film, we first cleaned a glass plate by sonication in ethanol (30 mins) and then acetone (30 mins), and dried at room temperature (25 °C). A homogeneous P(VDF-TrFE) solution was obtained by mixing 0.5 g of P(VDF-TrFE) powder in 5 mL of DMSO (0.1 g mL$^{-1}$), and the solution was stirred for 12 h with 300 rpm using magnetic mixer (C-MAG HS 4, IKA apparatus). This solution was subsequently dropped on the glass plate and dried at 70 °C for 7 h in a drying oven. After this, crystallinity treatment was conducted by incubating the film at 130 °C for 10 h in the vacuum drying oven(DZF-6020, Shanghai Jinghong). The P(VDF-TrFE) film was released from the glass after immersing in water and dried at 25 °C.

### P(VDF-TrFE)/PEDOT:PSS composite film

To ensure that the P(VDF-TrFE) film used show the same |$d_{33}$| values (-20 pC N$^{-1}$), the P(VDF-TrFE) film with a thickness of 80 μm was poled by corona poling at room temperature before fabricating the electrode. For the preparation of PEDOT:PSS solution with DMSO, 0.15 g PVP and 0.5 mL PEDOT:PSS were added to 10 mL DMSO, and stirred for 6 h with 600 rpm using the magnetic mixer. PVP was added to the PEDOT:PSS solution to change the contact angle and create an electrode because P(VDF-TrFE) films are hydrophobic. Various PEDOT:PSS solutions with DMSO (10 mL) (V$_{PEDOT:PSS}$/V$_{DMSO}$, from 5 vol%, 10 vol%, 15 vol%, 20 vol%, 25 vol% to 30 vol%) were prepared when PVP holds 0.15 g. The PEDOT:PSS solutions were dropped onto the P(VDF-TrFE) film and then placed in the oven at 45 °C for 30 min. To prepare the PEDOT:PSS solution with DMF, 0.15 g PVP and 0.5 mL PEDOT:PSS were added to 10 mL DMF, and stirred at 600 rpm for 6 h. For metal electrode devices, the poled P(VDF-TrFE) film was sputter-coated using a small ion sputterer (DM200, Hefei Dingshuo). The resistivity of the PEDOT:PSS electrode was measured using the four-point probe method (RTS-9, 4Probes Tech).

### Sensor device fabrication

To protect the sensors from damage due to mechanical excitation and water, it is necessary to encapsulate the P(VDF-TrFE)/PEDOT:PSS composite film in PDMS. For this purpose, the P(VDF-TrFE)/PEDOT:PSS composite film was encapsulated in PDMS (PDMS: Sylgard, 184 silicone elastomer) and curing agent (10:1 wt/wt) and dried at 45 °C for 30 min. For the sensor arrays, nine P(VDF-TrFE)/PEDOT:PSS composite films with a size of 10 × 10 mm were encapsulated in PDMS and curing agent and dried at 45 °C for 30 min (Fig. S1a).

### Device characterization

A spiral micrometer (MDC-25SX, Mitutoyo) was used to measure the thickness. The morphology and elemental composition of the composite film were analyzed using a combination of scanning electron microscopy (GeminiSEM 300, Carl Zeiss) and atomic force microscopy (MFP-3D infinity, Oxford Instruments). The SO$_3$ topography was visualized by TOF-SIMS (PHI nanoTOFII, Physical Electronics). The piezoelectric coefficient of the polarization P(VDF-TrFE) film after was measured using a quasistatic $d_{33}$ instrument (ZJ-3AN, IACAS) at room temperature. The measuring head of ZJ-3AN contains an electromagnetic force drive that generates a low-frequency (110 Hz) alternating force (0.5 N). Pressure tests of the devices were performed in the range of 0.025–100 kPa using a pressure-controlled motor at room temperature (PR-BDM8-100F, PURI Materials). Piezoelectric current was measured using a digital oscilloscope (TBS-2000B, Tektronix) with 1 MΩ input impedance and a low-noise current preamplifier (SR570, Stanford Research Systems) with 1 MΩ input impedance (Fig. S1b). Piezoelectric voltage was measured using an electrometer (6517B, Keithley) with an input impedance of 200 TΩ. Heating in the pyroelectric characterization experiments was induced by 808-nm near infrared (NIR) laser radiation (PSU-H-LED, Changchun new industries optoelectronics) (Fig. S1c). Here, temperature changes (0.05–10 °C) were monitored using a thermocouple and data acquisition modules (NI 9211, National Instruments), and the laser irradiation was modulated using a timed switching flap (GCI-73, Daheng Group). The pyroelectric voltage and current were collected using an electrometer (6514, Keithley) with an input impedance of 200 TΩ. Electrical signals from the sensor array were recorded using a multichannel capture device (USB 5630, Art Technology) with an input impedance of 10 MΩ. The capacitance of PEDOT:PSS/P(VDF-TrFE) was measured using a precision impedance analyzer (4294 A, Agilent). The polarization-electric loop was measured using TF ANALYZER 3000 (aixACCT).

### Simulation

The phase field method was applied to simulate the polarization of P(VDF-TrFE) with a cross-linked layer. The polarization of P(VDF-TrFE) can be obtained by solving the time-dependent Ginzburg-Landau equations

$$\frac{\partial P_i(\mathbf{r},t)}{\partial t} = -L\frac{\delta F}{\delta P_i(\mathbf{r},t)}\ (i=1,2,3) \tag{1}$$

where r is the spatial vector and $i$ = 1, 2, and 3 correspond to the $x$, $y$, and $z$ directions, respectively. $L$ is the kinetic coefficient, and $F$ is the

total free energy of the system. The total free energy per unit volume has the form of

$$F = \int_v f_{LD}(P_i) + f_G(P_{i,j}) + f_{elec}(P_i, E) \, dv \tag{2}$$

where $f_{LD}(P_i)$ is the Landau-Devonshire free energy density, $f_G(P_{i,j})$ is the gradient energy density, and $f_{elec}(P_i, E_i)$ is the electric energy density. The Landau free energy density is given by $f_{LD}(P) = \frac{\alpha}{2} P_3^2 + \frac{\beta}{4} P_3^4 + \frac{\gamma}{6} P_3^6$, where $\alpha = \alpha_0(T - T_0)$, $\beta$, $\gamma$ are the dielectric stiffness coefficient, $T$ is temperature. The lowest order of the gradient energy density is taken as $f_G(P_{i,j}) = \frac{K_1}{2}\left(\frac{\partial P_1}{\partial x}\right)^2 + \frac{K_2}{2}\left(\frac{\partial P_2}{\partial y}\right)^2 + \frac{K_3}{2}\left(\frac{\partial P_3}{\partial z}\right)^2$, where $K_1$, $K_2$ and $K_3$ are the gradient coefficients. The electrostatic energy density with the applied external electric field ($E_i$) is described as $f_{elec}(P_i, E_i) = \frac{1}{2}(E_1 P_1 + E_2 P_2 + E_3 P_3)$. The values of the parameters of the simulation were listed in Table S5.

A P(VDF-TrFE) film with a thickness of 32 nm was simulated, where the bottom and top layers represent the PEDOT:PSS electrodes. Discrete grid points of $64 \times 64 \times 32$ with a cell size of $\Delta x = \Delta y = \Delta z = 1$ nm was used. The finite difference method for spatial derivatives and the Runge–Kutta method of order four for temporal derivatives were employed to solve Eq. (1). The piezoelectric properties of the film can be obtained by $d_{33} = 2\varepsilon_{33}\varepsilon_0 \left[ Q_{11} - \frac{2s_{11}Q_{12}}{s_{11} + s_{12}} \right] P$, where $Q_{11}$ and $Q_{12}$ are the electrostrictive coefficients, $S_{11}$ and $S_{12}$ are the elastic compliances. $\varepsilon_{33}$ is the relative dielectric constant and has the form of $\varepsilon_{33} = 1 + \left[ \varepsilon_0(\alpha + 3\beta P_3^2 + 5\gamma P_3^4) \right]^{-1}$, where $\varepsilon_0$ is the dielectric constant of the vacuum. The pyroelectric coefficient has the form of $p = \frac{\partial P}{\partial T}$, where $P$ is the polarization.

## Reporting summary

Further information on research design is available in the Nature Portfolio Reporting Summary linked to this article.

## Data availability

All data are available in the supplementary information and in the figshare database with the link: https://doi.org/10.6084/m9.figshare.22666126.

## Code availability

The codes used for the simulations presented in this article are available upon request from the corresponding author.

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

## Acknowledgements

This work was supported by the Natural Science Foundation of Guang-dong province (2023A1515012638), Shenzhen Natural Science Funds for Distinguished Young Scholar (No. RCJC20210706091949018), the National Natural Science Foundation of China (No. 12274152, 12175191), and the initial financial support from HUST (No. 3004110155, Y.L.). We also acknowledge the in part support of the Guangdong Provincial Key Laboratory Program (No. 2021B1212040001) from the Department of Science and Technology of Guangdong Province. WSL acknowledges support from the Tencent Foundation through the XPLORER PRIZE.

## Author contributions

B.L. conceived this project. B.L., C.C, and B.Y. prepared the samples. B.Y., P.Z., B.D., Q.L., F.W., and C.C. conducted experimental. C.C, B.L., and B.Y. analyzed the data. B.L. performed phase field simulation. B.L. and C.C. prepared the manuscript. B.L. and Y.L. revised the manuscript. W.L., B.L., Y.L., and P.H. funding acquisition. W.L., supervision. All authors participated in the analysis and discussion.

## Competing interests

The authors declare no competing interests.
