## [Peer Review File · Nature Communications]

REVIEWER COMMENTS

Reviewer #1 (Remarks to the Author):

The authors reported interface-modified P(VDF-TrFE) films which present improve electroactivity. The results are interesting. However, current version is far away from acceptance for publication in Nat Commun. Some of my comments include:

1) details on film and device fabrication, how to build up the experimental system and how to do the experiments are missing. Some original experimental data are suggested to be provided at least in the supplementary file.

2) some experiments in both the main text and the supplementary file are much similar, such as pressure response measurements. the showing ordering of experimental results is a little disordered. The authors are suggested to re-arrange all these experimental results to better present their findings.

3) some supplementary figures are not introduced in the main text.

4) in Experimental section

PEDOT:PSS solution concentration, all equipment manufacturers, how to conduct poling process...

how about P(VDF-TrFE) thickness before PEDOT:PSS deposition, is it the same for all films used in this work?

'then transferred it to a hot air oven to dry the solvent for 30 minutes': at what temperature?

'by mixing 0.05-0.3 mL of DMSO and 0.15 g of PVP': why the authors use PVP? The authors did not provide any explanation in the whole manuscript.

'a 5% DMSO-doped PEDOT:PSS solution': what is the meaning of 5%?

'piezoelectric signals recorded using a low-noise current preamplifier (Stanford, SR570) and an oscilloscope (TBS-2000B).': the oscilloscope is used to recorded the output from the preamplifier? However, in all experimental data, the authors didn't provide any current data from the piezoelectric films.

Which equipment did the authors used for voltage measurement? How about its input impedance, since we know that P(VDF-TrFE) films usually have very high resistance.

'The pyroelectric voltage of the device was collected using an electrometer (KEITHLEY 6514)', 'multichannel capture device (USB5630)': again, how about their input impedance.

5) Fig.1 and relevant text:

There are two F images.

How to realize the 'flat interface' in Fig.1A?

How to conduct d_{33} measurement for the device 'without electrode' in Fig.1G?

The original experimental data are needed for the work in Fig.1G-1I to show how to get those d_{33} , S_p and S_t data.

How to define and obtain S_p and S_t coefficients?

How to get d_{33} coefficient? At what frequency and pressure?

Labels in Fig.1H and 1I are not clear.

What is the meaning of those different colored areas in Fig.1A-1C?

'The NII structure in the composite film is formed by the interweaving of these PEDOT:PSS clusters with P(VDF-TrFE) chains': what are 'clusters'? Any experimental results to prove the existence of these clusters?

'the typical three-layer thin film structure into a five-layer arrangement': Where are five layers? In my opinion, there only three layers. Here DMSO only roughened P(VDF-TrFE) surface, rather than result in new layers.

How did the authors conduct d_{33} and pyroelectric measurements (equipment, measurement system, pressure, frequency, temperature, original data...)?

How about film thickness? We know that thickness is also a key parameter to determine voltage output of piezoelectric devices.

'PEDOT:PSS electrode enables the composite film to absorb most of the radiant heat energy': Why? Any evidences?

6) Fig.2, Fig.S3 and relevant text

In Fig.2A and 2B: what is the meaning of polarization distribution? To what kind of polarization did the green, blue and red areas correspond? What is the difference of polarization distribution between Fig. A and B? Have both films in Fig. A and B been polarized? If so, both results should show uniform polarization distribution. If not, what is the meaning to simulate un-polarized films?

'Because DMSO can enhance the conductivity of PEDOT:PSS, more charge can be collected.': Why high conductivity PEDOT:PSS can result in larger d_{33} ? Why it is not due to different corrosion degree from DMSO and DMF, i.e. DMF has weaker corrosion to P(VDF-TrFE) which induces a little flat interface? Did the authors conduct cross-section SEM analysis on the composite films corroded by DMF?

Fig.2C: besides the increase of effective contact area, are there any other reasons that cause the increase of d_{33} ?

In Fig.2G, what is the device of 'Oum'? How to realize this film?

In Fig. E and F: are all data measured from experiments? Since PEDOT:PSS are conductive, why its thickness influences capacitance and d_{33} ?

'when the PEDOT:PSS electrode thickness increases, which is caused by the increase in the NII thickness': this conclusion is a little arbitrary, since many factors can influence NII thickness.

How to change PEDOT:PSS thickness (concentration, deposition times or deposition process)?

Fig.S3: it is a little arbitrary to separate each layer by SEM cross-section image. Furthermore, since PEDOT:PSS is conductive, it should correspond to the brightest area in SEM images. However, in Fig.S3 PEDOT:PSS areas are the darkest. Why?

'the piezoelectric coefficient is determined by the permittivity': the authors should give a brief introduction on how to calculate d_{33} through permittivity. How to set those parameters during calculation?

About Fig. 2D: the increase of capacitance should be due to the increase of effective contact area, rather than permittivity. For a specific material, its permittivity should be a constant.

In Fig.2E, F and H, the authors give PEDOT:PSS thickness dependence of capacitance and d_{33} . However, I don't think PEDOT:PSS thickness is a good parameter to directly determine capacitance and d_{33} values. It is NII thickness rather than PEDOT:PSS thickness that directly determine both values.

7) Fig.3, Fig. S4, Fig. S5 and relevant text:

Fig.S4B caption: 'Variation in the resistivity of the composite film as a function of the bending cycle.' Is it the resistance of the PEDOT:PSS electrode or the whole composite film? Since P(VDF-TrFE) is insulating, its resistance should be much higher. Furthermore, how to get this resistance value, is it square resistance or others?

What is 'coefficient of determination'? How to get it?

For all voltage measurements in Fig.3, S4 and S5, is the 'voltage' open-circuit voltage? Which equipment is used for this measurement? How about its input impedance?

In Fig.S5, how to separate the contribution of temperature and force on sensor's response, since both stimuli can induce electrical response of P(VDF-TrFE). For example in Fig.S5c, both temperature and force are simultaneous applied, how did the authors get pressure and temperature curves, separately?

In Fig.S5B, temperature curve in K with time (recorded by a temperature gauge) is suggested to be provided.

How did the authors conduct experiments in Fig. S5 A and B, since in Fig. S5A only pressure is applied, how did the authors measure the temperature? Similarly, in Fig. S5B, only temperature is changed, how did the authors measure the pressure?

What is the difference of 'held on the device for 5 s' and 'pressing is maintained for 5 s' in experimental operation?

In Fig.3A and 3D, the authors didn't indicate which results are from piezoelectric films with PEDOT:PSS or metal electrodes.

In Fig.3D, why both curves have different shape? What the difference of both curves? In the upper curve the increase of temperature results in the increase of voltage, however, in the lower curve, temperature increase induces voltage decrease, why?

In Fig.3D, the real temperature change (in K or °C) with time is also suggested to be provided.

In Fig. 3G and 3H, the curves are not well presented.

Which kind of composite device (thickness, NII, DMSO and so on) is used for the measurement in Fig.3?

8) Fig.4, Fig. S6-8 and relevant text

'Each waveform initially decreases the output voltage, indicating that the first syllable in both terms is pronounced similarly.' 'decrease' is not good to describe the curve. The conclusion is too arbitrary, since other different syllable can also result in similar waveform (downward curve).

In Fig.4c, how did the authors fix the sensor for pulse measurement? The three characteristic peaks for pulse signal are not clear. A comparison, for example, pulse signal detected by simply electrospun P(VDF-TrFE) fiber device [Mater. Chem. Front., 2021,5, 5679] is much clearer than this work, why? Furthermore, the shape of pulse signal is far different from those detected by piezoelectric effect[Mater. Chem. Front., 2021,5, 5679][Adv. Electron. Mater. 2022, 8, 2200012], why?

Again, how to separate the contribution of force and temperature?

'The temperature signals are similarly modified as the temperature radiated by the fingertip varies according to the magnitude of the force being exerted': why?

For Fig.4F, where did those data come from? How to conduct the measurements?

In Fig. S6 and S7, how to fabricate the (array) devices? Dimensions of sensing units? How to do the measurements?

The schematic diagram in Fig.S7D did not have an iron block.

9) the device for d33 measurement is encapsulated by PDMS? We know that substrate can also influence the piezoelectric output, since large deformation is expected for piezoelectric film deposited on flexible substrate. Similarly, thick PEDOT:PSS may also contribute like thick substrate, as may also induce large piezoelectric output. The authors are suggested to discuss about the contribution of flexible substrate.

Reviewer #2 (Remarks to the Author):

The work is novel in presenting a network interconnection interface (NII) route with polymer electrodes for boosting the sensitivity of ferroelectric polymer sensors. It should be noted that penetrated metal electrodes has been demonstrated, and I suggest the author emphasize the principal difference between previous reported penetrated electrodes strategy and NII in this paper. Please see my comments below.

Major comments

1. The author mentioned that “the heat diffusion associated with the radiation absorption coefficient of the metal electrodes decreases the magnitude of the pyroelectric voltage²⁸⁻³⁰.” However, Ref 30 is related to triboelectric nanogenerators. To demonstrate the advantage of NII route to penetrated metal electrodes, the pyroelectric performance should be discussed theoretically and experimentally in details.
2. Fig. 1G shows the piezoelectric coefficient (d_{33}) of the P(VDF-TrFE) film prepared with PEDOT:PSS and different metal electrodes. However, the d_{33} results of penetrated metal electrodes were not given. Similarly in Fig 1H, P(VDF-TrFE) is different from PVDF. I recommend deleting the result of PVDF.
3. The author mentioned that “In the calculation, the piezoelectric coefficient is determined by the permittivity⁴⁹. The calculated permittivity shows a monotonically increasing tendency as a function of NII thickness (Fig. 2D).” More theoretical details or formula in calculating the permittivity would be useful.
4. Figure 3 shows the result in high pressure range (larger than 1kPa). I strongly recommend including supplementary results that show the sensing performance in low pressure ranges such as 0~100Pa.
5. The time response of the sensor was not given. I recommend including the measurement results of time response of the sensor.
6. For discussion, very thick electrodes (>100 μ m) are required to achieve a high d_{33} as shown in Fig 2H. Compared to normal metal electrode with a thickness in the range of several hundred nanometers to several micrometers, it is very challenging to pattern the polymer electrodes using traditional wet or dry etching methods.

Minor comments

In the sentence of “The conventional poly(vinylidene fluoride-co-trifluoroethylene) copolymer (P(VDF-TrFE)) has a low sensor voltage output because of its relatively low piezoelectric and pyroelectric coefficients”. “P(VDF-TrFE)” -> “P(VDF-TrFE)”

Reviewer #3 (Remarks to the Author):

Paper on influencing a network interconnection interface to enhance properties. The novelty stems from the network/conductivity electrode.

“For a fixed polymer thickness, expanding the contact area between the polymer and electrodes causes the capacitance to increase” - does the effective thickness also change/decrease due to the conductive electrode?

Was a poling process applied to the device. This is not clear in the paper, and if not, how is the polarisation achieved? I also see no classical polarisation-electric field hysteresis loops? This would be desirable.

It is stated the d_{33} is measured "using a quasistatic d_{33} instrument (ZJ-3AN) - in the discussion on Figure 2 it is stated " the calculation, the piezoelectric coefficient is determined by the permittivity." - this could be more clear. It would be good if the d_{33} could be clearly described as the value is large >80 pC/N

Is there a danger of developing a short-circuit in the material?

"The cross-sectional area of the applied force determines the $|d_{33}|$ of P(VDF-TrFE)." - this is an unusual comment as most d_{33} meters apply a small area (almost point force) and collect charge - and apply force to the same surface as the charge is collected removes the effect of area. This links to my comment on the lack of clarity on how d_{33} was measured.

The application of the material at the end is interesting but not vital and innovative - the key is to demonstrate enhancement in materials properties.

List of Responses to the Reviewers' Comments

We would like to thank the reviewers for their valuable comments and suggestions on this manuscript, which greatly improves the quality of our manuscript. Following these comments and suggestions, we have made careful revisions to our previous manuscript (marked in red colour), and provide response to the comments (marked in blue colour) point-by-point as follows:

Responses to Reviewer #1 (Remarks to the Author):

The authors reported interface-modified P(VDF-TrFE) films which present improve electroactivity. The results are interesting. However, current version is far away from acceptance for publication in Nat. Commun. Some of my comments include:

We appreciate the reviewer's acknowledgement of our manuscript. We have carefully revised our previous manuscript and addressed all comments and concerns point-by-point.

Comments 1: details on film and device fabrication, how to build up the experimental system and how to do the experiments are missing. Some original experimental data are suggested to be provided at least in the supplementary file.

Response: Thanks for the reviewer's comments. The details of the preparation of the film and the device were added in the revised manuscript as shown below.

1) The details of the preparation of the film and equipment.

"P(VDF-TrFE) film synthesis: To fabricate a P(VDF-TrFE) film, we first cleaned a glass plate by sonication in ethanol (30 min) and then acetone (30 min), and dried at room temperature (25 °C). A homogeneous P(VDF-TrFE) solution was obtained by mixing 0.5 g of P(VDF-TrFE) powder in 5 mL of DMSO (0.1 g mL⁻¹), and the solution was stirred for 12 h with 300 rpm using magnetic mixer (C-MAG HS 4, IKA apparatus). This solution was subsequently dropped on the glass plate and dried at 70 °C for 7 h in a drying oven. After this, crystallinity treatment was conducted by incubating the film at 130 °C for 10 h in the vacuum drying oven (DZF-6020, Shanghai Jinghong). The P(VDF-TrFE) film was released from the glass after immersing in water and dried at 25 °C.

P(VDF-TrFE)/PEDOT:PSS composite film: To fabricate the P(VDF-TrFE)/PEDOT:PSS composite film, the P(VDF-TrFE) film with the thickness of 80 μm was first poled by corona poling at room temperature. For the preparation of PEDOT:PSS solution with DMSO, 0.15 g PVP and 0.5 mL PEDOT:PSS were added to 10 mL DMSO, and stirred for 6 h with 600 rpm using the magnetic mixer. PVP was added to the PEDOT:PSS solution to change the contact angle and create an electrode because P(VDF-TrFE) films are hydrophobic. Various PEDOT:PSS solutions with DMSO (10 mL) ($V_{\text{PEDOT:PSS}}/V_{\text{DMSO}}$, from 5 vol%, 10 vol%, 15 vol%, 20 vol%, 25 vol% to 30 vol%) were prepared when PVP holds 0.15 g. The PEDOT:PSS solutions were dropped onto the P(VDF-TrFE) film and then placed in the oven at 45 °C for 30 min. To prepare the PEDOT:PSS solution with DMF, 0.15 g PVP and 0.5 mL PEDOT:PSS were added to 10 mL DMF, and stirred at 600 rpm for 6 hours. For metal electrode devices, the P(VDF-TrFE) film was sputter-coated using a small ion sputterer (DM200, Hefei Dingshuo). The resistivity of the PEDOT:PSS electrode was measured using the four-point probe method (RTS-9, 4Probes Tech).

Sensor device fabrication: To protect the sensors from damage due to mechanical excitation and water, it is necessary to encapsulate the P(VDF-TrFE)/PEDOT:PSS composite film in PDMS. For this purpose, the P(VDF-TrFE)/PEDOT:PSS composite film was encapsulated in PDMS (PDMS: Sylgard, 184 silicone elastomer) and curing agent (10:1 wt/wt) and dried at 45 °C for 30 min. For the sensor arrays, nine P(VDF-TrFE)/PEDOT:PSS composite films with a size of 10 \times 10 mm were encapsulated in PDMS in the same way as described above.” (page 12-13)

2) The experimental details were added in the revised manuscript as follows. The experimental setup for the piezoelectric and pyroelectric measurements was added in the revised supporting information (Fig. S1).

“*Device characterization:* A spiral micrometer (MDC-25SX, Mitutoyo) was used to measure the thickness.scanning electron microscopy (GeminiSEM 300, Carl Zeiss) and atomic force microscopy (MFP-3D infinity, Oxford Instruments). The SO₃ topography was visualized by TOF-SIMS (PHI nanoTOFII, Physical Electronics).a quasistatic d_{33} instrument (ZJ-3AN, IACAS) at room temperature. The

measuring head of ZJ-3AN contains an electromagnetic force drive that generates a low-frequency (110 Hz) alternating force (0.5 N). Pressure tests of the devices were performed in the range of 0.07-100 kPa using a pressure-controlled motor at room temperature (PR-BDM8-100F, PURI Materials). Piezoelectric current was measured using a digital oscilloscope (TBS-2000B, Tektronix) with 1 M Ω input impedance and a low-noise current preamplifier (SR570, Stanford Research Systems) with 1 M Ω input impedance (Fig. S1B). Piezoelectric voltage was measured using an electrometer (6517B, Keithley) with an input impedance of 200 T Ω . Heating in the pyroelectric characterization experiments was induced by 808-nm near infrared (NIR) laser radiation (PSU-H-LED, Changchun new industries optoelectronics) (Fig. S1C). Here, temperature changes (0.05-10 $^{\circ}$ C) were monitored using a thermocouple and data acquisition modules (NI 9211, National Instruments), and the laser irradiation was modulated using a timed switching flap (GCI-73, Daheng Group). The pyroelectric voltage and current were collected using an electrometer (6514, Keithley) with an input impedance of 200 T Ω . Electrical signals from the sensor array were recorded using a multichannel capture device (USB 5630, Art Technology) with an input impedance of 10 M Ω . The capacitance of PEDOT:PSS/P(VDF-TrFE) was measured using a precision impedance analyzer (4294A, Agilent). The polarization-electric loop was measured using TF ANALYZER 3000 (aixACCT).” (page 13-14)

Fig. S1 (A) Schematic diagram of the P(VDF-TrFE)/PEDOT:PSS composition film preparation and sensor device fabrication. The setup of (B) pyroelectric and (C) piezoelectric output testing system.

3) The original experimental data were added in the revised supporting information as follows.

“Table S1. The d_{33} of P(VDF-TrFE) films without and with different electrodes.

Table S2. Measured data used to calculate the pyroelectric coefficient in the P(VDF-TrFE)/PEDOT:PSS composite film.

Table S3. The d_{33} the P(VDF-TrFE) films with PEDOT:PSS electrode as a function of NII thickness.

Table S4. The parameter values used in the simulation.

Table S5. Piezoelectric and pyroelectric voltage (peak-to-peak) as a function of pressure and temperature.”

Comment 2: 2) some experiments in both the main text and the supplementary file are much similar, such as pressure response measurements. the showing ordering of experimental results is a little disordered. The authors are suggested to re-arrange all these experimental results to better present their findings.

Response: Thanks for the reviewer’s suggestion. To address this comment, we rearrange the experimental and simulation results as follows.

1) Schematic diagrams of the preparation of the P(VDF-TrFE)/PEDOT:PSS composition and sensor device were combined into a new figure (Fig. S1A). The setup of the pyroelectric and piezoelectric output test system was added in the revised supporting manuscript as Fig. S1B and S1C, respectively.

2) The pressure and temperature sensitivity shown in Fig. 1 of the first version of our manuscript was added to Fig. 3 of the revised manuscript as Fig. 3H and 3I. The corresponding description and discussion have also been revised.

3) The calculated and experimentally measured pyroelectric coefficient was added to Fig. 2 of the revised manuscript. Two new figures of polarization electric hysteresis loops and X-ray diffraction of P(VDF-TrFE) were included in the revised manuscript.

4) In the revised manuscript, a new application of the thumb pulse test sensor was added in Fig. 4.

Comment 3: 3) some supplementary figures are not introduced in the main text.

Response: Thanks for the reviewer's careful check. According to the reviewer's suggestion, the supplementary figures of S1 and S2 in the submitted supporting information were added in the revised manuscript.

“A schematic diagram of the P(VDF-TrFE)/PEDOT:PSS composite film and the corresponding test system can be found in Fig. S1.” (page 4)

“In the P(VDF-TrFE)/PEDOT:PSS composite film with NII, the charges generated by pressure and temperature are trapped inside the NII in addition to those accumulated on the surface of the composite film. The electromechanical and electrothermal energy conversion is enhanced when more charges accumulate (Fig. S4A). In contrast, charges in conventional P(VDF-TrFE) devices with metal electrodes accumulate only on the surfaces (Fig. S4B).” (page 4)

Comment 4: In Experimental section

4) PEDOT:PSS solution concentration, all equipment manufacturers, how to conduct poling process...

Response: Thanks for the reviewer's comment. In the revised manuscript, details of equipment (type and manufacturer) and experiments were added, as shown previously.

Comment 5: how about P(VDF-TrFE) thickness before PEDOT:PSS deposition, is it the same for all films used in this work?

Response: Thanks for the reviewer's comment. In this work, all the P(VDF-TrFE) film has the same thickness (80 μm) before PEDOT:PSS solutions deposition as follows.

“To fabricate the P(VDF-TrFE)/PEDOT:PSS composite film, the P(VDF-TrFE) film with the thickness of 80 μm was first poled by corona poling at room temperature.” (page 12)

Comment 6: ‘then transferred it to a hot air oven to dry the solvent for 30 minutes’: at what temperature?

Response: Thanks for the reviewer's comment. According to the reviewer's suggestion, the temperature used was supplemented in the revised manuscript as follows.

“The PEDOT:PSS solutions were dropped onto the P(VDF-TrFE) film and then placed in the oven at 45 $^{\circ}\text{C}$ for 30 min.” (page 13)

Comment 7: ‘by mixing 0.05-0.3 mL of DMSO and 0.15 g of PVP’: why the authors use PVP? The authors did not provide any explanation in the whole manuscript. ‘a 5% DMSO-doped PEDOT:PSS solution’: what is the meaning of 5%?

Response: Thanks for the reviewer’s careful check. The information on the concentration of the DMSO- and DMF-doped PEDOT:PSS solution was supplemented in the revised manuscript as follows.

“For the preparation of PEDOT:PSS solution with DMSO, 0.15 g PVP and 0.5 mL PEDOT:PSS were added to 10 mL DMSO, and stirred for 6 h with 600 rpm using the magnetic mixer. PVP was added to the PEDOT:PSS solution to change the contact angle and create an electrode because P(VDF-TrFE) films are hydrophobic. Various PEDOT:PSS solutions with DMSO (10 mL) ($V_{\text{PEDOT:PSS}}/V_{\text{DMSO}}$, from 5 vol%, 10 vol%, 15 vol%, 20 vol%, 25 vol% to 30 vol%) were prepared when PVP holds 0.15 g. The PEDOT:PSS solutions were dropped onto the P(VDF-TrFE) film and then placed in the oven at 45 °C for 30 min. To prepare the PEDOT:PSS solution with DMF, 0.15 g PVP and 0.5 mL PEDOT:PSS were added to 10 mL DMF, and stirred at 600 rpm for 6 h.”
(page 12-13)

Comment 8: ‘piezoelectric signals recorded using a low-noise current preamplifier (Stanford, SR570) and an oscilloscope (TBS-2000B).’: the oscilloscope is used to recorded the output from the preamplifier? However, in all experimental data, the authors didn’t provide any current data from the piezoelectric films.

Response: The current data of pressure and temperature monitor were provided in Fig S10 as show in below.

“Piezoelectric current was measured using a digital oscilloscope (TBS-2000B, Tektronix) with 1 M Ω input impedance and a low-noise current preamplifier (SR570, Stanford Research Systems) with 1 M Ω input impedance (Fig. S1B).”

“Fig. S10 (A) Piezoelectric and (B) pyroelectric current of the composite film as a function of the pressure and temperature, respectively.”

Comment 9: Which equipment did the authors use for voltage measurement? How about its input impedance, since we know that P(VDF-TrFE) films usually have very high resistance. ‘The pyroelectric voltage of the device was collected using an electrometer (KEITHLEY 6514)’, ‘multichannel capture device (USB5630)’: again, how about their input impedance.

Response: Thanks for the reviewer’s comments. According to the reviewer’s suggestion, the equipment information and the input impedance were added in the revised manuscript as follows.

“Piezoelectric voltage was measured using an electrometer (6517B, Keithley) with an input impedance of 200 TΩ.” (page 13)

“The pyroelectric voltage and current were collected using an electrometer (6514, Keithley) with an input impedance of 200 TΩ.” (page 14)

“Electrical signals from the sensor array were recorded using a multichannel capture device (USB 5630, Art Technology) with an input impedance of 10 MΩ.” (page 14)

Comment 10: Fig.1 and relevant text:

Response: Thanks for the reviewer’s comments. According to the reviewer’s suggestion, Fig. 1 was revised as shown follows.

“Fig. 1. Formation of the NII structure in a P(VDF-TrFE)/PEDOT:PSS composite film and its piezoelectric and pyroelectric properties. Schematic illustration of the formation of the network interconnection interface in the P(VDF-TrFE)/PEDOT:PSS composite film. (A) P(VDF-TrFE)/PEDOT:PSS composite film **normal** interface. (B) DMSO promotes the interdiffusion process. (C) Schematic diagram of the formed NII structure. (D) and (E) SEM images of the cross-section of the P(VDF-TrFE)/PEDOT:PSS composite film and energy dispersive spectrum of fluorine. (F) Two-dimensional distribution of SO₃ in the interfacial components obtained by TOF-SIMS. (G) Piezoelectric coefficient of P(VDF-TrFE) with different electrode materials. **Summary of the (H) piezoelectric coefficient and (I) pyroelectric coefficient.**” (page 5-6)

Comment 11: There are two F images. How to realize the ‘flat interface’ in Fig.1A?

Response: Thanks for the reviewer’s careful check. The letter F in Fig. 1E stands for the element fluorine. In the revised manuscript, F in Fig. 1E has been changed to fluorine and SO₃ in Fig. 1F was replaced by sulfur trioxide. The flat interface used in Fig. 1A indicated the interface when the PEDOT:PSS solution dropped on the P(VDF-TrFE) without interconnect. The flat interface in Fig. 1A was revised as normal interface.

To address this comment, the following revision was made in the revised manuscript:

1) In the revised manuscript, “F” in Fig 1E has been renamed “Fluorine”, and “flat interface” has been renamed “Normal interface”

“**Fig. 1** (A) P(VDF-TrFE)/PEDOT:PSS composite film **normal** interface.” (page 6)

2) “Then, **sulfur trioxide (SO₃)** was chosen as the marker of PEDOT:PSS to examine the formation of the NII layer by time-of-flight secondary ion mass spectrometry (TOF-SIMS).” (page 4)

Comment 12: How to conduct d_{33}

measurement for the device ‘without electrode’ in Fig.1G?

Response: For d_{33} measurement using a quasistatic d_{33} instrument (ZJ-3AN, IACAS), electrode is not necessary. Therefore, without electrode here means that neat copolymer film was used for the d_{33} measurement. This result provides a reference to show the effect of different electrodes on d_{33} values.

To address this comment, we add one sentence “**Without electrode indicates that no electrodes were used for d_{33} measurement to provide a reference to show the effect of different electrodes on d_{33} values.**” (page 4)

Comment 13: The original experimental data are needed for the work in Fig.1G-1I to show how to get those d_{33} , S_p and S_t data.

Response: Thanks for the reviewer’s suggestion. To address this comment, the original data d_{33} of the P(VDF-TrFE) films without and with various electrode were added in revised supporting information (Table. S1, Table. S2), and the original data of pyroelectric and piezoelectric voltage were added as Table S5.

“**Table S1. The d_{33} of P(VDF-TrFE) films without and with different electrodes.**”

“**Table S2. Measured data used to calculate the pyroelectric coefficient in the P(VDF-TrFE)/PEDOT:PSS composite film.**”

“**Table S5. Piezoelectric and pyroelectric voltage (peak-to-peak) as a function of pressure and temperature.**”

Comment 14: How to define and obtain S_p and S_t coefficients?

Response: To address this comment, the definition of S_p and S_t in the revised supporting information has been given as follows.

“*Calculation of piezoelectric and pyroelectric voltage output of the devices*”

The piezoelectric voltage output of the devices (S_p) can be obtained from the equation of $S_p = \Delta V_p / \Delta P$, where ΔV_p and ΔP are the relative change of piezoelectric voltage output and applied pressure, respectively¹. The pyroelectric voltage output of the devices $S_t = \Delta V_t / \Delta T$, where ΔV_t and ΔT are the relative change of pyroelectric voltage output and temperature, respectively.”

Comment 15: How to get d33 coefficient? At what frequency and pressure?

Response: To address this comment, the detail information about the d_{33} test were added in the revised manuscript as follows. “The piezoelectric coefficient of the polarization P(VDF-TrFE) film after was measured using a quasistatic d_{33} instrument (ZJ-3AN, IACAS) at room temperature. The measuring head of ZJ-3AN contains an electromagnetic force drive that generates a low-frequency (110 Hz) alternating force (0.5 N).” (page 13)

Comment 16: Labels in Fig. 1H and 1I are not clear.

Response: Thanks for the reviewer’s comment. To address this comment, the labels in Fig. 1H and 1I were revised to make it more clearly in revised manuscript as follows.

“Fig. 1Summary of the (H) piezoelectric coefficient and (I) pyroelectric coefficient.” (page 5-6)

Comment 17: What is the meaning of those different colored areas in Fig.1A-1C?

Response: Thanks for the reviewer’s careful check. The blue color represents PEDOT:PSS, and the two colors (origin and brown) represent P(VDF-TrFE). The image of SEM (Fig. 1D) shows that the non-uniform cross-section of the P(VDF-TrFE) film has high and low areas, resulting in the PEDOT:PSS filling the low areas. To address this comment, the meaning of color representation was added in Fig. 1A-1C as follows.

“Fig. 1. Formation of the NII structure in a P(VDF-TrFE)/PEDOT:PSS composite film and its piezoelectric and pyroelectric properties.” (page 5)

Comment 18: ‘The NII structure in the composite film is formed by the interweaving of these PEDOT:PSS clusters with P(VDF-TrFE) chains’: what are ‘clusters’? Any experimental results to prove the existence of these clusters?

Response: The PEDOT:PSS clusters in this paper denote the PEDOT:PSS-rich room. We assume that the NII structure in the composite film consists of a P(VDF-TrFE)-rich room and a PEDOT:PSS-rich room, as shown in the cross section image of the PEDOT:PSS/P(VDF-TrFE) composite film below. The red circle represents the PEDOT:PSS-rich room, and the white circle demotes the P(VDF-TrFE)-rich room.

To address this comment, the sentence was modified as **“Dissolution of PEDOT:PSS in DMSO leads to the formation of PEDOT:PSS clusters (Fig. S2A). The NII structure in the composite film is formed by the interweaving of these PEDOT:PSS clusters with P(VDF-TrFE) chains.”** (page 4)

“Fig. S2A SEM image of the cross-section of the PEDOT:PSS/P(VDF-TrFE) composite film.”

Comment 19: ‘the typical three-layer thin film structure into a five-layer arrangement’: Where are five layers? In my opinion, there only three layers. Here DMSO only roughened P(VDF-TrFE) surface, rather than result in new layers.

Response: Thanks for the reviewer’s comment. To clarify whether two layers are formed, we designed the following experiment. A drop of DMSO solution is applied to the central region of polarized P(VDF-TrFE) and dried to form an electrode. The cross

section of the PEDOT:PSS/P(VDF-TrFE) composite film with and without the electrode is observed using SEM see below. It was found that the smoothness of the surface and the thickness of the P(VDF-TrFE) film did not change. Therefore, we agree with the reviewer's statement that the composite film has a three-layer structure.

Fig. R1 The cross-section of P(VDF-TrFE)/PEDOT:PSS composite film

To address this comment, we changed five layers to three layers. “The formation of an NII structure do not changes the typical three-layer thin film structure.” (page 4)

Comment 20: How did the authors conduct d_{33} and pyroelectric measurements (equipment, measurement system, pressure, frequency, temperature, original data...)?

Response: Thanks for the reviewer's comment. To address this comment, detailed information on the d_{33} and pyroelectric measurements has been added in the revised manuscript and in the supporting information as follows.

Detailed information on the d_{33} measurements: “The piezoelectric coefficient of the polarization P(VDF-TrFE) film after was measured using a quasistatic d_{33} instrument (ZJ-3AN, IACAS) at room temperature. The measuring head of ZJ-3AN contains an electromagnetic force drive that generates a low-frequency (110 Hz) alternating force (0.5 N).” (page 13)

Detailed information on the pyroelectric measurements: “Heating in the pyroelectric characterization experiments was induced by 808-nm near infrared (NIR) laser radiation (PSU-H-LED, Changchun new industries optoelectronics) (Fig. S1C). Here, temperature changes (0.05-10 °C) were monitored using a thermocouple and data acquisition modules (NI 9211, National Instruments), and the laser irradiation was modulated using a timed switching flap (GCI-73, Daheng Group). The pyroelectric voltage and current were collected using an electrometer (6514, Keithley) with an input impedance of 200 TΩ. Electrical signals from the sensor array were recorded using a

multichannel capture device (USB 5630, Art Technology) with an input impedance of 10 MΩ.” (page 14)

“The pyroelectric coefficient of composite film obtains for experimental

The pyroelectric coefficient of the composite film can be determined by the following equation:

$$p = \frac{I}{A \cdot dT / dt} \quad (1)$$

where I is the current, A is the area of electrode, T is the temperature and t is the time. Measured data used to calculate the pyroelectric coefficient in the P(VDF-TrFE)/PEDOT:PSS composite film was listed in Table. S2.”

The corresponding original data were added in the revised supporting information (Table S1 and S5).

“Table S1. The d_{33} of P(VDF-TrFE) films without and with different electrodes”

“Table S5. Piezoelectric and pyroelectric voltage (peak-to-peak) as a function of pressure and temperature.”

Comment 21: How about film thickness? We know that thickness is also a key parameter to determine voltage output of piezoelectric devices.

Response: Thanks for the reviewer’s comment. The voltage of the piezoelectric devices depends on the thickness of the P(VDF-TrFE) film, and a high voltage can be achieved with a large thickness. Therefore, the thickness of the P(VDF-TrFE) film used was 80 μm. To address this comment, we add one sentence:

“To fabricate the P(VDF-TrFE)/PEDOT:PSS composite film, the P(VDF-TrFE) film with the thickness of 80 μm was first poled by corona poling at room temperature.” (page 12)

Comment 22: ‘PEDOT:PSS electrode enables the composite film to absorb most of the radiant heat energy’: Why? Any evidences?

Response: The absorption spectra showed that the PEDOT:PSS electrode has high absorption (92%) in the near infrared region compared to the Au electrode (8%), as follows. The following discussion on why the PEDOT:PSS electrode can absorb more

thermal radiation energy compared to the Au electrode was added in the revised manuscript.

“The PEDOT:PSS electrode allows the composite film to absorb more radiant heat energy compared to the Au electrode due to the higher absorption performance^{45,46}. Subsequently, the pyroelectric response was drastically improved by increasing the derivative of temperature with respect to time (dT/dt).” (page 5)

Fig. R2 Absorption spectrum of (A) PEDOT:PSS electrode⁴⁹, (B) Au electrode⁵⁰.

Comment 23: 6) Fig.2, Fig.S3 and relevant text

Response: Thanks for the reviewer’s comments. To address this comment, Fig. 2 was revised as follows.

“**Fig. 2 Mechanisms of the NII in performance improvement.** (A) Calculated and experimentally measured piezoelectric coefficient and (B) pyroelectric coefficient of composite films at different NII thickness ratios. (C) Variation in the capacitance of the composite P(VDF-TrFE)/PEDOT:PSS film with the PEDOT:PSS electrode thickness. Inset: Comparison of the capacitance of a P(VDF-TrFE) film with metal electrodes with

that of the composite film. (D) Piezoelectric coefficient of the composite film versus the thickness of the NII with a DMSO volume fraction of 5 vol%. (E) Piezoelectric coefficient of the composite film as a function of the DMSO volume fraction. (F) Comparison of d_{33} in the composite film with DMSO and DMF solution.” (page 8)

Comment 24: In Fig.2A and 2B: what is the meaning of polarization distribution?

Response: Polarization distribution denotes the magnitude of polarization in the P(VDF-TrFE) film with different area. In the revised manuscript, a color bar has been added to indicate the extent of polarization (see below). The calculated and experimentally measured pyroelectric coefficients of composite films with different NII thickness ratios were added to Fig. 2 as Fig. 2B. Therefore, the Fig. 2A and 2B were moved from the submitted manuscript to the supporting information as Fig. S5.

To address this comment, we add one sentence to the revised manuscript as follows:

“Fig. S5 shows the simulated polarization distribution of the polarized P(VDF-TrFE) films without and with NII, respectively, where the color indicates the magnitude and direction of polarization.” (page 6)

Fig. S5 Phase field simulation of the polarization distribution of P(VDF-TrFE) in the P(VDF-TrFE)/PEDOT:PSS structure (A) without and (B) with the NII structure.

Comment 25: To what kind of polarization did the green, blue and red areas correspond?

Response: The green, blue, and red regions indicate the direction of polarization in the P(VDF-TrFE) film. In the red region, the magnitude of polarization is 0.11 C m^{-2} , indicating the direction of polarization along the z -direction according to the color bar. The blue areas indicate the polarization direction with the negative polarization (-0.11 C m^{-2}) along the negative z -direction.

To address this comment, the following sentence was included in the revised manuscript: “Fig. S5 shows the simulated polarization distribution of the polarized

P(VDF-TrFE) films without and with NII, respectively, where the color indicates the magnitude and direction of polarization.” (page 6)

Comment 26: What is the difference of polarization distribution between Fig. A and B? Have both films in Fig. A and B been polarized? If so, both results should show uniform polarization distribution. If not, what is the meaning to simulate un-polarized films?

Response: In the simulation, the total energy of the films (F) is expressed as a function of the polarization and has the form of $F = \int_V f_{LD}(P_i) + f_G(P_{i,j}) + f_{elec}(P_i, E_i) dv$, where $f_{LD}(P_i)$ is the Landau-Devonshire free energy density, $f_G(P_{i,j})$ is the gradient energy density, and $f_{elec}(P_i, E_i)$ is the electric energy density. Therefore, the simulated P(VDF-TrFE) films were polarized.

The direction of polarization of ferroelectric materials tends to coincide with the direction of the external electric field after polarization. The grain boundaries, defects, and dislocations in ferroelectric materials would prevent the reversal of polarization direction, resulting in the polarization direction of ferroelectric materials usually not being uniformly distributed and referred to as multidomain structure¹⁻³.

The differences in polarization distribution between Fig. 2A and 2B were shown below. The proposal draws our attention to the fact that the distinction of the category structure is not the focus of the discussion. Therefore, the pyroelectric and piezoelectric coefficients were calculated and added in the revised manuscript as Fig. 2A and 2B. The domain structure of P(VDF-TrFE) with or without PEDOT:PSS was included in the revised supporting manuscript.

Fig. R3 The difference of polarization distribution between Fig. S5A and S5B

References

- 1 Damjanovic, D. Logarithmic frequency dependence of the piezoelectric effect due to pinning of ferroelectric-ferroelastic domain walls. *Physical Review B* **55**,

R649-R652, (1997)

- 2 Scott, J. F. & Dawber, M. Oxygen-vacancy ordering as a fatigue mechanism in perovskite ferroelectrics. *Applied Physics Letters* **76**, 3801-3803, (2000)
- 3 Kontsos, A. & Landis, C. M. Computational modeling of domain wall interactions with dislocations in ferroelectric crystals. *International Journal of Solids and Structures* **46**, 1491-1498 (2009)

To address this comment, the following sentence was included in the revised manuscript: “Fig. S5 shows the simulated polarization distribution of the polarized P(VDF-TrFE) films without and with NII, respectively, where the color indicates the magnitude and direction of polarization.” (page 6)

Comment 27: ‘Because DMSO can enhance the conductivity of PEDOT:PSS, more charge can be collected.’: Why it is not due to different corrosion degree from DMSO and DMF, i.e. DMF has weaker corrosion to P(VDF-TrFE) which induces a little flat interface? Did the authors conduct cross-section SEM analysis on the composite films corroded by DMF?

Response: Thanks for the reviewer’s suggestion. To clarify the corrosion of DMSO and DMF on the P(VDF-TrFE) film, the cross-sectional images SEM of the composite films were examined with different solutions under the same conditions (0.11 mL at 40 °C for 30 min), as shown in Fig. S7. It can be seen that the thickness of the NII obtained from DMSO is larger than that obtained from DMF. This can be explained by the fact that the boiling point of DMSO (190 °C) is higher than that of DMF (155 °C). The PEDOT:PSS/DMSO solution would remain longer on the P(VDF-TrFE) film and lead to DMSO having a higher corrosion of P(VDF-TrFE) compared to DMF. This could be another reason for the larger d_{33} in the PEDOT:PSS/P(VDF-TrFE) composite film.

To address this comment, the following sentence was included in the revised manuscript: “In addition, the high boiling point of DMSO (190 °C) leads to higher corrosion of P(VDF-TrFE) compared to DMF (155 °C), as shown in Fig. S9.” (page 7)

“Fig. S9 Cross-sectional SEM images of PEDOT:PSS/P(VDF-TrFE) composite films

prepared using (A) DMSO and (B) DMF solution.”

Comment 28: Fig.2C: besides the increase of effective contact area, are there any other reasons that cause the increase of d_{33} ?

Response: Thanks for the reviewer’s inspiring comment. The following discussion was added in the revised supporting information as follow.

“Analysis of the increase in the piezoelectric coefficient (d_{33}) attributable to NII

Phase content: the d_{33} of the P(VDF-TrFE) film depends on its β -phase content. For the P(VDF-TrFE) and P(VDF-TrFE)/PEDOT:PSS composite film, X-ray diffraction shows that the strongest diffraction at $2\theta=20.5^\circ$ of β -phase with (110)/(200) planes, as shown in Fig. S2B. The Bragg diffraction intensity of P(VDF-TrFE) is greater than that of the P(VDF-TrFE)/PEDOT:PSS composite film, indicating that the β -phase phase content does not increase.

2) PEDOT:PSS electrode: the d_{33} of the PEDOT:PSS electrode without P(VDF-TrFE) film was measured with ZJ-3AN, and therefore the PEDOT:PSS electrode does not contribute to the improvement of the d_{33} .

3) Thickness: the d_{33} of the P(VDF-TrFE) film depends on its thickness, and the P(VDF-TrFE) film used in this work holds $80\ \mu\text{m}$. The thickness of the P(VDF-TrFE) film does not contribute to the d_{33} .”

Comment 29: In Fig.2G, what is the device of ‘0um’? How to realize this film?

Response: The legend in Fig. 2G indicates the thickness of the total PEDOT:PSS thickness, $0\ \mu\text{m}$ indicates that no PEDOT:PSS is present. To clarify this, the legend was changed to indicate the total thickness of the PEDOT:PSS/P(VDF-TrFE) composite film. In this work, the P(VDF-TrFE) films with the thickness of $80\ \mu\text{m}$ were used, therefore the legend was $80, 130, 180$ and $230\ \mu\text{m}$.

“Fig. 2....(E) Piezoelectric coefficient of the composite film as a function of the DMSO volume fraction.” (page 8)

Comment 30: In Fig. E and F: are all data measured from experiments? Since PEDOT:PSS are conductive, why its thickness influences capacitance and d_{33} ?

Response: Thanks for the reviewer's comment. To address this comment, the equipment for capacity measurement was added in the revised manuscript.

“The capacitance of PEDOT:PSS/P(VDF-TrFE) was measured using a precision impedance analyzer (4294A, Agilent).” (page 14)

The statement about the influence of PEDOT:PSS on capacity and d_{33} was added in the revised manuscript as follows.

“According to the Koop's model⁴⁸, the dielectric heterogeneous network conductivity can be considered as a conducting PEDOT:PSS separated by the high-resistance P(VDF-TrFE). The space charge created by the stress due to the piezoelectric effect builds up at the P(VDF-TrFE), controlling the available free charge carriers at the P(VDF-TrFE) and leading to a higher dielectric constant.” (page 6)

Comment 31: ‘when the PEDOT:PSS electrode thickness increases, which is caused by the increase in the NII thickness’: this conclusion is a little arbitrary, since many factors can influence NII thickness.

Response: Thanks for the reviewer's comment. It is noted that the formation of NII is caused by the dissolution of the P(VDF-TrFE) film by the DMSO in the PEDOT:PSS solution and leads to the interweaving of PEDOT:PSS with P(VDF-TrFE). The NII thickness would increase when the volume of the applied PEDOT:PSS solution increases, as shown in Fig. S7. To address this comment, the following sentences were included in the revised manuscript:

“Then, the effect of PEDOT:PSS solution on NII thickness was investigated. Fig. S7 shows the NII thickness as a function of PEDOT:PSS thickness with PEDOT:PSS volume fraction of 5 vol% which is obtained from a SEM image of the cross-section of the P(VDF-TrFE)/PEDOT:PSS composite film. The NII thickness increases when the PEDOT:PSS thickness increases, which is controlled by the volume of the applied PEDOT:PSS solution and the DMSO volume fraction.” (page 7)

In addition, the influence of DMOS and DMF on the NII thickness of DMOS were also present. Therefore, we add “In addition, the high boiling point of DMSO (190 °C) leads

to higher corrosion of P(VDF-TrFE) with greater NII thickness compared to DMF (155 °C), as shown in Fig. S9.” into our revised manuscript (page 7)

Comment 32: How to change PEDOT:PSS thickness (concentration, deposition times or deposition process)?

Response: In the experimental, the PEDOT:PSS thickness was controlled by the PEDOT:PSS volume fraction (from $V_{\text{PEDOT:PSS}}/V_{\text{DMSO}}$, from 5 vol%, 10 vol%, 15 vol%, 20 vol%, 25 vol% to 30 vol%) and PEDOT:PSS volume (from 0.12, 0.14, 0.16, 0.18 to 0.2 mL).

To address this comment, we add “The NII thickness increases when the PEDOT:PSS thickness increases, which is controlled by the volume of the applied PEDOT:PSS solution and the DMSO volume fraction.” into our revised manuscript. (page 7)

Comment 33: Fig.S3: it is a little arbitrary to separate each layer by SEM cross-section image. Furthermore, since PEDOT:PSS is conductive, it should correspond to the brightest area in SEM images. However, in Fig.S3 PEDOT:PSS areas are the darkest. Why?

Response: Fig. 1D and 1E proved that the NII can be separated by the lightness from cross-sectional SEM images of PEDOT:PSS/P(VDF-TrFE) composite film. The NII was darker than the P(VDF-TrFE). The cross-sectional SEM images of PEDOT:PSS/P(VDF-TrFE) composite film with different PEDOT:PSS electrode thicknesses were retested and added in the revised manuscript as Fig. S6 (see below).

“Fig. S7 Cross-sectional SEM images of PEDOT:PSS/P(VDF-TrFE) composite film with different PEDOT:PSS thicknesses: (A) 5 μm, (B) 50 μm, (C) 100 μm, and (D) 125 μm. (E) NII thickness as a function of PEDOT:PSS thickness.”

Due to the uneven sectioning of the PEDOT:PSS electrode, some areas are darker and some areas are lighter (Fig. S2A).

“Fig. S2 (A) SEM image of the cross-section of the PEDOT:PSS/P(VDF-TrFE) composite film. (B) X-ray diffraction of P(VDF-TrFE) with and without PEDOT:PSS”

To address this comment, we add “It has been proved that the NII thickness can be obtained for the brightness of the cross-section of the composite film where the NII is darker than the P(VDF-TrFE).” into our revised manuscript. (page 4)

Comment 34: ‘the piezoelectric coefficient is determined by the permittivity’: the authors should give a brief introduction on how to calculate d_{33} through permittivity. How to set those parameters during calculation?

Response: Calculation details for d_{33} , dielectric constant, and pyroelectric were added in the revised manuscript. The parameters used for the calculation were listed in the revised supporting information (Table S4). To address this comment, the following revision was made in the revised manuscript:

“The piezoelectric properties of the film can be obtained by

$$d_{33} = 2\varepsilon_{33}\varepsilon_0 \left[Q_{11} - \frac{2s_{11}Q_{12}}{s_{11} + s_{12}} \right] P, \text{ where } Q_{11} \text{ and } Q_{12} \text{ are the electrostrictive coefficients,}$$

S_{11} and S_{12} are the elastic compliances. The pyroelectric coefficient has the form of

$$p = \left| \frac{\partial P}{\partial T} \right|, \text{ where } P \text{ is the polarization.}” \text{ (page 15)}$$

“Table S4. The parameter values used in the simulation.”

Parameter	Value	Parameter	Value

a_0	$7.5 \times 10^7 \text{ J m C}^{-2} \text{ K}$	G_{11}	$9.96 \times 10^{-10} \text{ N m}^4 \text{ C}^{-2}$
b	$-1.9 \times 10^{12} \text{ J m}^5 \text{ C}^{-4} \text{ K}$	G_{12}	0
g	$1.9 \times 10^{14} \text{ J m}^9 \text{ C}^{-6} \text{ K}$	G_{44}	$4.98 \times 10^{-10} \text{ N m}^4 \text{ C}^{-2}$
S_{11}	$3.32 \times 10^{-10} \text{ m}^2 \text{ N}^{-1}$	Q_{11}	$0 \text{ m}^4 \text{ C}^{-2}$
S_{12}	$-1.44 \times 10^{-10} \text{ m}^2 \text{ N}^{-1}$	Q_{12}	$3 \text{ m}^4 \text{ C}^{-2}$
ε_0	$8.85 \times 10^{-10} \text{ F m}^{-1}$	T_0	307 K

Comment 35: About Fig. 2D: the increase of capacitance should be due to the increase of effective contact area, rather than permittivity. For a specific material, its permittivity should be a constant.

Response: Thanks for the reviewer's comment. The dielectric is a material property that does not depend on the geometrical characteristics. In the composite film, the effective contact area between the P(VDF-TrFE) and the PEDOT:PSS increases due to the NII and forms a heterogeneous network conductivity. According to Koop's model, the dielectric heterogeneous network conductivity can be considered as a conducting PEDOT:PSS separated by the high-resistance P(VDF-TrFE). The space charge induced by the stress due to the piezoelectric effect builds up at the P(VDF-TrFE), controlling the available free charge carriers at the P(VDF-TrFE) and increase the dielectric constant. To address this comment, the following revision was made in the revised manuscript:

“According to the Koop's model⁴⁸, the dielectric heterogeneous network conductivity can be considered as a conducting PEDOT:PSS separated by the high-resistance P(VDF-TrFE). The space charge created by the stress due to the piezoelectric effect builds up at the P(VDF-TrFE), controlling the available free charge carriers at the P(VDF-TrFE) and leading to a higher dielectric constant.” (page 6)

Comment 36: In Fig.2E, F and H, the authors give PEDOT:PSS thickness dependence of capacitance and d33. However, I don't think PEDOT:PSS thickness is a good parameter to directly determine capacitance and d33 values. It is NII thickness rather than PEDOT:PSS thickness that directly determine both values.

Response: Thanks for the reviewer's inspiring comment. According to the reviewer's suggestion, we measured the cross-section of the PEDOT:PSS/P(VDF-TrFE)

composite film with different PEDOT:PSS thickness to determine the relationship between the NII thickness and the PEDOT:PSS thickness. The NII thickness increases with increasing PEDOT:PSS thickness. Figs. 2E, F and H were revised as the dependence of NII thickness on capacitance and d_{33} . A corresponding description and new figure (Fig. S6E) were added in the revised manuscript and supplemental materials (see below).

“Fig. S7 Cross-sectional SEM images of PEDOT:PSS/P(VDF-TrFE) composite film with different PEDOT:PSS thicknesses: (A) 5 μm, (B) 50 μm, (C) 100 μm, and (D) 125 μm. (E) NII thickness as a function of PEDOT:PSS thickness.”

Comment 37: 7) Fig.3, Fig. S4, Fig. S5 and relevant text:

Response: Thanks for the reviewer’s comment. According to the reviewer’s suggestion, the Fig. 3 was revised as follows.

“Fig. 3 Piezoelectric and pyroelectric voltage output of PEDOT:PSS/P(VDF-TrFE) with the NII. (A) Response of the composite film (top) and P(VDF-TrFE) with Au electrode (bottom) to periodic pressing and release at 100 kPa. (B) Response of the composite film to dynamic loading with pressures in the range of 0.07–100 kPa. (C) Variation in the sensor output voltage with pressure. (D) Response of composite film (top) and P(VDF-TrFE) with Au electrode (bottom) to periodic heating and cooling at 10 K. (E) Response of the composite film to periodic heating with temperature increases in the 0.05–10 K range. (F) Variation in the sensor output voltage with temperature. Summary of (G) piezoelectric and (H) pyroelectric sensitivity. (I) The long-term stability of the piezoelectric and pyroelectric responses of the composite film in response to an applied pressure of 100 kPa (top) and the pyroelectric response to a temperature of 10 K (bottom).” (page 10)

Comment 38: Fig.S4B caption: 'Variation in the resistivity of the composite film as a function of the bending cycle.' Is it the resistance of the PEDOT:PSS electrode or the whole composite film? Since P(VDF-TrFE) is insulating, its resistance should be much higher. Furthermore, how to get this resistance value, is it square resistance or others?

Response: Thanks for the comment by the reviewer. The resistance of P(VDF-TrFE) is generally much higher than that of PEDOT:PSS because P(VDF-TrFE) is insulating and PEDOT:PSS is conducting. The resistance shown in Fig. S4B was the square resistance of the PEDOT:PSS electrode.

To clarify this comment, we add device and its type in the revised manuscript, and the radius of curvature during the bending cycle was added in the revised supporting information.

“The resistivity of the PEDOT:PSS electrode was measured using the four-point probe method (RTS-9, 4Probes Tech).” (page 13)

“Fig. S11 (A) Home made bending test system. (B) Variation in the resistivity of the composite film as a function of the bending cycle at a radius of curvature ≤ 5.3 mm.”

Comment 39: What is ‘coefficient of determination’? How to get it?

Response: We add more discussion of the coefficient of determination and the method to deduce this coefficient.

In the revised manuscript: “The pressure sensitivity (S_p), defined as the slope of the graph in Fig. 3C (see supporting information), is 2.2 V kPa⁻¹, and the coefficient of determination (R_{pp}^2 , see the supporting information) for the pressure sensitivity was determined to be 0.9989.” (page 9)

In the revised supporting information: “ R^2 in the manuscript indicates the degree of fit of the trend line indicator, and its numerical magnitude reflects the degree of fit between the estimated value of the trend line and the corresponding actual data. The higher the degree of fit, the more reliable the trend line is. The R-squared value is calculated as follows. The R-squared value is calculated as follows^{2,3}:

$$R^2 = \frac{SSR}{SST} = \frac{\sum_{i=1}^n (\hat{y}_i - \bar{y})^2}{\sum_{i=1}^n (y_i - \bar{y})^2}$$

where \hat{y}_i is the voltage, and \bar{y} is the average voltage.”

Comment 40: For all voltage measurements in Fig.3, S4 and S5, is the ‘voltage’ open-circuit voltage? Which equipment is used for this measurement? How about its input impedance?

Response: All voltage measurements in Fig. 3, S4 and S5 are the open circuit voltage, because the voltage is much larger than the current. Piezoelectric and pyroelectric voltage signals were measured using different devices.

We clarify these comments by adding the following sentences into our manuscript.

“Piezoelectric voltage was measured using an electrometer (6517B, Keithley) with an input impedance of 200 TΩ.” (page 14)

“The pyroelectric voltage and current were collected using an electrometer (6514, Keithley) with an input impedance of 200 TΩ.” (page 14)

Comment 41: In Fig. S5, how to separate the contribution of temperature and force on sensor’s response, since both stimuli can induce electrical response of P(VDF-TrFE).

For example in Fig. S5c, both temperature and force are simultaneous applied, how did the authors get pressure and temperature curves, separately?

Response: To address this comment, we add the following discussion in the revised supporting information:

“How to separate the contribution of piezoelectric and pyroelectric effects in the pressure and temperature and stimuli

1) Voltage was measured using a bottle of water at room temperature (25 °C). The temperature of the water in the bottle was measured using a thermocouple to ensure that it was the same as the ambient temperature and that no pyroelectric voltage was generated. Therefore, it can be assumed that this voltage is generated only by the piezoelectric effect (U_{pi}).

2) The water in the bottle was heated to different temperatures (usually 30, 35 and 40 °C), and then the voltage was measured using the 6514. In this case, both the piezoelectric and pyroelectric effects contribute to the measured voltage (U_{to}). The contribution of the pyroelectric voltage (U_{py}) to the total voltage can be determined by $U_{py}=U_{to}-U_{pi}$. Then a proportionality factor ($P_{pi\text{py}}$) between the pyroelectric and piezoelectric effect is defined as $P_{pi\text{py}} = U_{pi} : U_{py}$.

3) If the devices were used to measure temperature and pressure simultaneously, the contribution of the pyroelectric effect and the piezoelectric effect to the voltage can be determined by $P_{pi\text{py}}$.

4) The piezoelectric and piezoelectric voltage curves show that the response time of these two effects is different. This feature has been found to be helpful to show the pyroelectric effect and the piezoelectric effect.

It should be noted that the third coefficient, related to the piezoelectric effect, results from the temperature gradient along the polar axis of the ferroelectrics⁴. The piezoelectric stress can contribute to the pyroelectric voltage when the sample is larger. In this work, pyroelectric is mainly considered as a temperature monitoring application. We have assumed that the influence of piezoelectric effect on pyroelectric voltage has little effect on temperature monitoring.”

Comment 42: In Fig. S5B, temperature curve in K with time (recorded by a

temperature gauge) is suggested to be provided.

Response: To address this comment, the temperature curve was measured as a function of time using thermocouples and data acquisition modules (NI 9211, National Instruments). The temperature curve was included in Fig. S5A and S5B, which are shown as Fig. S12A and S12B in the revised manuscript as follows.

“Fig. S12 (A) Single-mode response of the sensor to contact with a glass bottle. (B) Single-mode response of the sensor to hot air (contactless heating). (C) Dual-mode response of the sensor to instantaneous finger contact. (D) Dual-mode response of the sensor to different modes of static contact: (up) “press-hold-release” and (down) “press-pressing-release”. In the former, a finger is pressed on the device, held for 5 s, and then released. In the latter, a finger is pressed on the device, pressure is maintained for 5 s, and the finger is released. The hold and pressure phases of finger contact produce different pyroelectric and piezoelectric signals”

Comment 43: How did the authors conduct experiments in Fig. S5 A and B, since in Fig. S5A only pressure is applied, how did the authors measure the temperature? Similarly, in Fig. S5B, only temperature is changed, how did the authors measure the pressure?

Response: During the loading/unloading experiments, a thermocouple is attached to the prepared sensor to measure the temperature. The temperature curve in Fig. S5A and S5B in the submitted manuscript is shown below as previously shown. During the pressing and releasing process, there is no temperature change and thus no pyroelectric voltage (Fig. S12A in the revised supporting information). The hot air pressure was

measured using the mechanical cyclic deformation system for flexible electronics (PR-BDM8-100F, PURI Materials) and no pressure can be measured. Therefore, there is no piezoelectric voltage. To address this comment, we add temperature curve to Fig. S12A and S12B as follows.

“Fig. S12 (A) Single-mode response of the sensor to contact with a glass bottle. (B) Single-mode response of the sensor to hot air (contactless heating). (C) Dual-mode response of the sensor to instantaneous finger contact. (D) Dual-mode response of the sensor to different modes of static contact: (up) “press-hold-release” and (down) “press-pressing-release”. In the former, a finger is pressed on the device, held for 5 s, and then released. In the latter, a finger is pressed on the device, pressure is maintained for 5 s, and the finger is released. The hold and pressure phases of finger contact produce different pyroelectric and piezoelectric signals.”

Comment 44: What is the difference of ‘held on the device for 5 s’ and ‘pressing is maintained for 5 s’ in experimental operation?

Response: In the ‘held on the device for 5 s’, the finger only touches the device and tries not to apply force. In the ‘pressing is maintained for 5 s’, a continuous force is applied when the finger comes into contact with the device. The test was designed to verify that our devices are suitable for complex stress/temperature conditions. The following description of these two processes were added in the revised supporting information. To address this comment, the following revision was made in the revised supporting manuscript:

“Fig. S12(D) Dual-mode response of the sensor to different modes of static contact: (up) “press-hold-release” and (down) “press-pressing-release”. In the former, a finger is pressed on the device, held for 5 s, and then released. In the latter, a finger is pressed on the device, pressure is maintained for 5 s, and the finger is released. The hold and pressure phases of finger contact produce different pyroelectric and piezoelectric signals.”

Comment 45: In Fig.3A and 3D, the authors didn’t indicate which results are from piezoelectric films with PEDOT:PSS or metal electrodes.

Response: Thanks for the reviewer’s careful check. To address this comment, we have added to the description in the revised manuscript. We have also revised the captions of Figs. 3A and 3D as follows.

“Fig. 3(A) Response of the composite film (top) and P(VDF-TrFE) with Au electrode (bottom) to periodic pressing and release at 100 kPa. (B) Response of the composite film to dynamic loading with pressures in the range of 0.07–100 kPa. (C) Variation in the sensor output voltage with pressure. (D) Response of composite film (top) and P(VDF-TrFE) with Au electrode (bottom).....” (page 10)

Comment 46: In Fig. 3D, why both curves have different shape? What the difference of both curves? In the upper curve the increase of temperature results in the increase of voltage, however, in the lower curve, temperature increase induces voltage decrease, why?

Response: The differences in thermal conductivity between the metal and PEDOT:PSS electrodes are the reason for the different shape of the pyroelectric voltage curve in Fig. 3D. A high thermal conductivity is important for a fast response of the pyroelectric detector¹, and leading to the difference shape of pyroelectric voltage in P(VDF-TrFE) with metal and PEDOT:PSS electrode.

According to the mechanism of the pyroelectric effect in P(VDF-TrFE), a voltage would be induced by temperature. The rise or fall of the voltage is determined by the positive or negative d_{33} on the side of P(VDF-TrFE) under the temperature. In the P(VDF-TrFE) film, the d_{33} on one side is positive, and the d_{33} on the back side of the film is negative. When the temperature applied on the P(VDF-TrFE) film where d_{33} is

positive, the stress would increase, while the temperature would cause the voltage decrease when d_{33} is negative. To address this comment, Fig. 3D has been revised to ensure that the output voltages can be compared as shown previously.

References

1 Crisman, E. E., Derov, J. S., Drehman, A. J. & Gregory, O. J. Large pyroelectric response from reactively sputtered aluminum nitride thin films. *Electrochemical and Solid-State Letters* **8**, H31, (2005).

Comment 47: In Fig.3D, the real temperature change (in K or oC) with time is also suggested to be provided.

Response: Thanks for the reviewer’s comment. According to the reviewer’s suggestion, the real temperature change (in °C) in Fig. 3D as a function of time was added in revised manuscript as shown previously.

Comment 48: In Fig. 3G and 3H, the curves are not well presented.

Response: Thanks for the reviewer's careful consideration. To address this comment, we have revised Fig. 3G and 3H to show the axis labels and values more clearly as shown formerly. In the revised manuscript, Fig. 3G and 3H were merged and renamed as Fig. 3I.

“Fig. 3..... (I) The long-term stability of the piezoelectric and pyroelectric responses of the composite film in response to an applied pressure of 100 kPa (top) and the pyroelectric response to a temperature of 10 K (bottom).” (page 10)

Comment 49: Which kind of composite device (thickness, NII, DMSO and so on) is used for the measurement in Fig.3?

Response: Thanks for the reviewer’s comments. To address this comment, we add more discussion on composite device:

“The piezoelectric and pyroelectric signal of the P(VDF-TrFE)/PEDOT:PSS film was studied with a total thickness of 230 μm and $d_{33}=86 \text{ pC N}^{-1}$, and the solvent of PEDOT:PSS was DMSO.” (page 8)

Comment 50: 8) Fig.4, Fig. S6-8 and relevant text

Response: Thanks for the reviewer’s comments. To address this comment, the Fig. 4 was revised as follows.

“**Fig. 4 Application of the P(VDF-TrFE)/PEDOT:PSS composite film as an ultrasensitive sensor.** A Pulse waveform from sensor monitoring of a thumb (top) and subject’s wrist (bottom). B Magnified view of one cycle of the electrical signal. C Speech recognition based on monitoring laryngeal activity while speaking different words.” (page 11)

Comment 51: ‘Each waveform initially decreases the output voltage, indicating that the first syllable in both terms is pronounced similarly.’ ‘decrease’ is not good to describe the curve. The conclusion is too arbitrary, since other different syllable can also result in similar waveform (downward curve).

Response: Thanks for the reviewer’s comments. The purpose of Fig. 4A is to verify that devices based on the composite film are able to detect the tiny pronunciation variations. It is true that the different syllables may result in different waveforms. To address this comment, we have made new discussions to show that our devices can be used to recognize speech patterns: “The prepared sensor was attached to the participant’s throat to test the possibility of devices as a word recognition system. Vibration of the throat was detected when different words were spoken. A single peak of output voltage was present when the monosyllabic word “Hi” was spoken, whereas the multisyllabic word “Hello” elicited a multimodal shape (Fig. 4C).” (page 11)

Comment 52: In Fig.4c, how did the authors fix the sensor for pulse measurement? The three characteristic peaks for pulse signal are not clear. A comparison, for example, pulse signal detected by simply electrospun P(VDF-TrFE) fiber device [Mater. Chem. Front., 2021,5, 5679] is much clearer than this work, why? Furthermore, the shape of pulse signal is far different from those detected by piezoelectric effect[Mater. Chem. Front., 2021,5, 5679][Adv. Electron. Mater. 2022, 8, 2200012], why?

Response: The difference in wrist pulse between our work and others (Mater. Chem. Front., 2021,5, 5679)[Adv. Electron. Mater. 2022, 8, 2200012) may be caused by the human body. The pulse at the wrist of the different participants were re-measured, as shown in Fig. 4A. It can be seen that the tidal, dicrotic, and percussion waves of the different individuals are different. The differences between this work and the other works were added in the revised manuscript as follows.

“Next, the fabricated real-time physiological monitoring device was placed on an adult human to demonstrate the potential and usability of the composite film in health monitoring. The device was attached to the thumbs up or wrist of various volunteers to detect subtle variation in arterial blood pressure (Fig. 4A). Frequency and amplitude of pulses are accurately reproduced in terms of distance between adjacent peaks and average peak amplitude in real time. Normal heart rate remains at 72 and 79 beats per minute (bpm) for different volunteers. The measured values are close to the performance of a commercially available device (inset in Fig. 4A), demonstrating the ability of the prepared device based on a composite film for real-time physiological monitoring. The detailed waveforms in Fig. 4B show characteristic peaks corresponding to percussion waves (p-waves), tidal waves (t-waves), and dicrotic waves (d-waves) in the human pulse, consistent with other work^{61,62.}” (page 11)

Comment 53: Again, how to separate the contribution of force and temperature?

Response: To address this comment, we add the following discussion in the revised supporting information as shown previously.

“How to separate the contribution of piezoelectric and pyroelectric effects in the pressure and temperature stimuli.....”

Comment 54: ‘The temperature signals are similarly modified as the temperature radiated by the fingertip varies according to the magnitude of the force being exerted’: why?

Response: When the external force applied to the object increases, the contact area between the sensor and the object increases, increasing the heat exchange per unit time and decreasing the interface thermal resistance due to the elasticity of the rubber glove and skin. As a result, dT/dt increases and leads to an increase in pyroelectric voltage.

To address this comment, the following revision was made in the revised supporting manuscript:

“When the external force applied to the object increases, the contact area between the sensor and the object also increases, increasing the heat exchange per unit time and decreasing the interface thermal resistance due to the elasticity of the rubber glove and skin. As a result, dT/dt increases and leads to an increase in pyroelectric voltage and make the voltage increase with increase of pressure, as shown in Fig. S13C.”

Comment 55: For Fig.4F, where did those data come from? How to conduct the measurements?

Response: The pyroelectric and piezoelectric voltages of Fig. 4F in the submitted manuscript were obtained using two 6514 (Keithley) with an input impedance of 200 TΩ. To address this comment, the following revision was made in the revised supporting information: “Fig. S13C shows the response of the sensor attached to the subject’s fingertip when objects with different weights were picked up. Both the pressure voltage (obtained from 6517B) and temperature voltage (obtained from 6514) generated by the device vary according to the object being carried.”

Comment 56: In Fig. S6 and S7, how to fabricate the (array) devices? Dimensions of sensing units? How to do the measurements?

Response: Thanks for the reviewer’s comments. To address this comment, the fabrication and measurement process of the array were added in the revised manuscript.

“*Sensor device fabrication:* To protect the sensors from damage due to mechanical excitation and water, it is necessary to encapsulate the P(VDF-TrFE)/PEDOT:PSS composite film in PDMS. For this purpose, the P(VDF-TrFE)/PEDOT:PSS composite

film was encapsulated in PDMS (PDMS: Sylgard, 184 silicone elastomer) and curing agent (10:1 wt/wt) and dried at 45 °C for 30 min. For the sensor arrays, nine P(VDF-TrFE)/PEDOT:PSS composite films with a size of 10×10 mm were encapsulated in PDMS in the same way as described above (Fig. S1A).” (page 13)

When the sensor array working, the voltage of 9 devices is need to be collect at the same time.

“Electrical signals from the sensor array were recorded using a multichannel capture device (USB 5630, Art Technology) with an input impedance of 10 MΩ.” (page 14)

Comment 57: The schematic diagram in Fig.S7D did not have an iron block.

Response: Thanks for the reviewer's careful check. An iron block has been added to the schematic diagram in Fig. S7D. This figure has been renamed Fig. S14C because some new figures were added in the supporting information.

“Fig. S14 Mapping images of the sensor system in contact with (A) an iron block, (B) water, and (C) fingers.”

Comment 58: 9) the device for d33 measurement is encapsulated by PDMS? We know that substrate can also influence the piezoelectric output, since large deformation is expected for piezoelectric film deposited on flexible substrate. Similarly, thick PEDOT:PSS may also contribute like thick substrate, as may also induce large piezoelectric output. The authors are suggested to discuss about the contribution of flexible substrate.

Response: Thanks for the reviewer’s inspiring comment. To address this comment, the effect of a flexible substrate on piezoelectric voltage were discussed and included in the revised supporting information: “The PDMS has no piezoelectric properties and cannot induce voltage under the pressure. When a stress applied to our devices encapsulated with PDMS, the deformation can be enlarged and increase the piezoelectric voltage. In this work, the thickness of the PDMS was 200 μm , and it can be considered that it has no influence on the piezoelectric voltage. The PEDOT:PSS was an electrode that also has no piezoelectric effect and increases the voltage.”

Responses to Reviewer #2 (Remarks to the Author):

The work is novel in presenting a network interconnection interface (NII) route with polymer electrodes for boosting the sensitivity of ferroelectric polymer sensors. It should be noted that penetrated metal electrodes has been demonstrated, and I suggest the author emphasize the principal difference between previous reported penetrated electrodes strategy and NII in this paper. Please see my comments below.

We appreciate the reviewer’s acknowledgement of our work. All questions and concerns were addressed point-by-point.

According to the reviewer’s suggestion, the main differences between the previously reported strategy of penetrating electrodes and the NII in this work were added in the revised manuscript as follows: “This piezoelectric current of NII-derived penetrating electrode (~ 76 nA) is larger than the previously reported penetrating electrodes (~ 20 nA) at 5 kPa²⁶, indicates it has high piezoelectric properties.” (page 9)

Major comments

Comment 1: The author mentioned that “the heat diffusion associated with the radiation absorption coefficient of the metal electrodes decreases the magnitude of the pyroelectric voltage²⁸⁻³⁰.” However, Ref 30 is related to triboelectric nanogenerators. To demonstrate the advantage of NII route to penetrated metal electrodes, the pyroelectric performance should discussed theoretically and experimentally in details.

Response: Thanks for the reviewer’s careful check. According to the reviewer suggestion, the pyroelectric the reference 30 were replaced by the following references.

“³⁰ Hsiao, C.-C. & Siao, A.-S. A High aspect ratio micropattern in freestanding bulk pyroelectric cells. *Energy Technol.* **6**, 883-898 (2018).”#

The theoretical and experimental analysis of the influence of NII on pyroelectric properties was supplemented in the revised manuscript as follows.

Experimentally analysis:

In the revised manuscript: “The calculated piezoelectric coefficient and pyroelectric properties increase with increasing NII thickness ratio, which is in accord with our experimental results (Fig. 2A)” (page 6)

In the revised supporting information,

“The pyroelectric coefficient of composite film obtains for experimental

The pyroelectric coefficient of the composite film can be determined by the following equation:

$$p = \frac{I}{A \cdot dT / dt} \quad (1)$$

where I is the current, A is the area of electrode, T is the temperature and t is the time. Measured data used to calculate the pyroelectric coefficient in the P(VDF-TrFE)/PEDOT:PSS composite film was listed in Table. S2.”

Table S2. Measured data used to calculate the pyroelectric coefficient in the P(VDF-TrFE)/PEDOT:PSS composite film.

p ($\mu\text{C m}^{-2}\text{K}^{-1}$)	I (nA)	dT/dt	A (cm^2)
27.8	1.8	2.59	1.44
59.6	4.46	0.52	1.44
67.5	7.1	0.73	1.44
70.4	23.2	2.29	1.44
94.7	60	4.4	1.44

Theoretically analysis:

The calculation details about the pyroelectric coefficient were added in the revised manuscript as follows.

“The pyroelectric coefficient has the form of $p = \left| \frac{\partial P}{\partial T} \right|$, where P is the polarization.”

(page 15)

“Fig. 2 Calculated and experimentally measured (A) piezoelectric coefficient and (B) pyroelectric coefficient of composite films at different NII thickness ratios.....”

(page 8)

Comment 2: Fig. 1G shows the piezoelectric coefficient (d_{33}) of the P(VDF-TrFE) film prepared with PEDOT:PSS and different metal electrodes. However, the d_{33} results of penetrated metal electrodes were not given. Similarly in Fig 1H, P(VDF-TrFE) is different from PVDF. I recommend deleting the result of PVDF.

Response: Thanks for the reviewer’s correction.

To address this comment, the cross section of the P(VDF-TrFE) with Au electrode was measured as follows. It shows that the metal electrode was not penetrated into the P(VDF-TrFE) in this work. Therefore, the d_{33} results of penetrated metal electrodes were not given.

Fig. R1 The cross section of the P(VDF-TrFE) with metal electrode.

To address this comment, all references in Fig. 1G and 1H were replaced by the P(VDF-TrFE)-related work as follows. In addition, the pressure and temperature

sensitivity in Fig. 1G and 1H in the submitted manuscript were re-arranged as Fig. 3H and 3I, respectively, in the revised manuscript as follows.

“Fig 1.....Summary of the (H) piezoelectric coefficient and (I) pyroelectric coefficient”

Fig 1H and 1I related references:

³⁴ Bhavanasi, V., Kusuma, D. Y. & Lee, P. S. Polarization orientation, piezoelectricity, and energy harvesting performance of ferroelectric PVDF-TrFE nanotubess synthesized by nanoconfinement. *Adv. Energy Mater.* **4**, 1400723 (2014).

³⁵ Yuan, X., Gao, X., Shen, X., Yang, J., Li, Z., & Dong, S. A 3D-printed, alternatively tilt-polarized PVDF-TrFE polymer with enhanced piezoelectric effect for self-powered sensor application. *Nano Energy* **85**, 105985 (2021).

³⁶ Lv, F., et al. In-situ electrostatic field regulating the recrystallization behavior of P(VDF-TrFE) films with high β-phase content and enhanced piezoelectric properties towards flexible wireless biosensing device applications. *Nano Energy* **100**,107507 (2022).

³⁷ Chai, Bin, et al. Modulus-modulated all-organic core-shell nanofiber with remarkable piezoelectricity for energy harvesting and condition monitoring. *Nano Lett.* **23**, 1810-1819 (2023).

³⁸ Shepelin, Nick A., et al. Interfacial piezoelectric polarization locking in printable Ti₃C₂T_x MXene-fluoropolymer composites. *Nat. Commun.* **12**, 3171 (2021).” (page 18)

.....

⁴⁰ Mahdi, R. I., Gan, W. C., Abd Majid, W. H., Mukri, N. I. & Furukawa, T. Ferroelectric polarization and pyroelectric activity of functionalized P(VDF-TrFE) thin film lead free nanocomposites. *Polym.* **141**, 184-193 (2018).

⁴¹ Mahdi, R. I., Gan, W. C. & Abd. Majid, W. H. Hot plate annealing at a low temperature of a thin ferroelectric P(VDF-TrFE) film with an improved crystalline structure for sensors and actuators. *Sens.* **14**, 19115-19127 (2014).

⁴² Kim, J. et al. High-performance piezoelectric, pyroelectric, and triboelectric nanogenerators based on P(VDF-TrFE) with controlled crystallinity and dipole alignment. *Adv. Funct. Mater.* **27**, 1700702 (2017).

⁴³ Aliane, A., Benwadih, M., Bouthinon, B., Coppard, R., Domingues-Dos Santos, F. & Daami, A. Impact of crystallization on ferro-, piezo- and pyro-electric characteristics in thin film P(VDF-TrFE). *Org. Electron.* **25**, 92-98 (2015).

⁴⁴ Luo, W. B., et al. Enhanced pyroelectric property of PMN-PT/P[VDF-TrFE] thick film by

optimizing poling temperature. *J. Mater. Sci. Mater. Electron.* **29**, 271-276 (2018).” (page 18)

“Fig.3... Summary of (G) piezoelectric and (H) pyroelectric sensitivity.”

Fig 3G and 1H related references:

- 39 Simate, A., Tondou, B., Souères, P. & Bergaud, C. Hybrid PVDF/PVDF-graft-PEGMA membranes for improved interface strength and lifetime of PEDOT:PSS/PVDF/ionic liquid actuators. *ACS Appl. Mater. Interfaces* **7**, 19966-19977 (2015).
- 40 Mahdi, R. I., Gan, W. C., Abd Majid, W. H., Mukri, N. I. & Furukawa, T. Ferroelectric polarization and pyroelectric activity of functionalized P(VDF-TrFE) thin film lead free nanocomposites. *Polym.* **141**, 184-193 (2018).
- 41 Mahdi, R. I., Gan, W. C. & Abd. Majid, W. H. Hot plate annealing at a low temperature of a thin ferroelectric P(VDF-TrFE) film with an improved crystalline structure for sensors and actuators. *Sens.* **14**, 19115-19127 (2014).
-
- 50 Qamar, Z. et al. Reinforcement of electroactive characteristics in polyvinylidene fluoride electrospun nanofibers by intercalation of multi-walled carbon nanotubes. *J. Polym. Res.* **24**, 1-9 (2017).
- 51 Hu, X. et al. Improved piezoelectric sensing performance of P(VDF-TrFE) nanofibers by utilizing BTO nanoparticles and penetrated electrodes. *ACS Appl. Mater. Interfaces* **11**, 7379-7386 (2019).
- 52 Yildirim, E. et al. A theoretical mechanistic study on electrical conductivity enhancement of DMSO treated PEDOT: PSS. *J. Mater. Chem. C* **6**, 5122-5131 (2018).
- 53 Cruz-Cruz, I., Reyes-Reyes, M., Aguilar-Frutis, M. A., Rodriguez, A. G. & López-Sandoval, R. Study of the effect of DMSO concentration on the thickness of the PSS insulating barrier in PEDOT:PSS thin films. *Synthetic Met.* **160**, 1501-1506 (2010).
- 54 Chen, X., Shao, J., Li, X. & Tian, H. A flexible piezoelectric-pyroelectric hybrid nanogenerator based on P(VDF-TrFE) nanowire array. *IEEE T. Nanotechnol.* **15**, 295-302 (2016).
- 55 An, S., Jo, H. S., Li, G., Samuel, E., Yoon, S. S. & Yarin, A. L. Sustainable nanotextured wave energy harvester based on ferroelectric fatigue-free and flexoelectricity-enhanced piezoelectric P(VDF-TrFE) nanofibers with BaSrTiO₃ nanoparticles. *Adv. Funct. Mater.* **30**, 2001150 (2020).
- 56 Wang, X., Yang, B., Liu, J., Zhu, Y., Yang, C. & He, Q. A flexible triboelectric-piezoelectric hybrid nanogenerator based on P(VDF-TrFE) nanofibers and PDMS/MWCNT for wearable devices. *Sci. Rep.* **6**, 36409 (2016).”(page 19)

Comment 3: The author mentioned that “In the calculation, the piezoelectric coefficient is determined by the permittivity⁴⁹. The calculated permittivity shows a monotonically increasing tendency as a function of NII thickness (Fig. 2D).” More theoretical details or formula in calculating the permittivity would be useful.

Response: Thanks for the reviewer's suggestion. Following the reviewer's suggestion, the calculation details for d_{33} , the pyroelectric coefficient, and the permittivity were added in the revised manuscript and revised supporting information. In addition, the parameters used in the simulation were added as Table S4 in the revised supporting information.

“The piezoelectric properties of the film can be obtained by

$$d_{33} = 2\varepsilon_{33}\varepsilon_0 \left[Q_{11} - \frac{2s_{11}Q_{12}}{s_{11} + s_{12}} \right] P, \text{ where } Q_{11} \text{ and } Q_{12} \text{ are the electrostrictive coefficients,}$$

S_{11} and S_{12} are the elastic compliances. ε_{33} is the relative dielectric constant and has the form of $\varepsilon_{33} = 1 + [\varepsilon_0(\alpha + 3\beta P^2 + 5\gamma P^4)]^{-1}$, where ε_0 is the dielectric constant of the

vacuum. The pyroelectric coefficient has the form of $p = \left| \frac{\partial P}{\partial T} \right|$, where P is the

polarization.” (page 15)

Comment 4: Figure 3 show the result in high pressure range (larger than 1 kPa). I strongly recommend including supplementary results that show the sensing performance in low pressure ranges such as 0~100 Pa.

Response: Thanks for the reviewer's comments. To address this comment, a pressure test equipment was brought for PURI Materials (PR-BDM8-100F) and the piezoelectric voltages were measured under a pressure of 70 Pa, which is added to Fig. 3 as follows.

“Pressure tests of the devices were performed in the range of 0.07-100 kPa using a pressure-controlled motor at room temperature (PR-BDM8-100F, PURI Materials).”

(page 13)

“Fig. 3 (B) Response of the composite film to dynamic loading with pressures in the range of 0.07–100 kPa.” (page 10)

Comment 5: The time response of the sensor was not given. I recommend including the measurement results of time response of the sensor.

Response: Thanks for the reviewer's comments. To address this comment, the response time of the composite film was added to the revised the manuscript as follows.

“The piezoelectric response time of the composite film (0.06 S) is shorter than that of the mental electrode (0.1 S), indicating that the composite film has better response properties.” (page 8)

“The pyroelectric response time of the composite film (1.77 S) is shorter than that of the mental electrode (3.80 S) because the PEDOT:PSS electrode can absorb more radiant heat energy and increase the dT/dt .” (page 9)

“Fig.3(A) Response of the composite film (top) and P(VDF-TrFE) with Au electrode (bottom) to periodic pressing and release at 100 kPa. (B) Response of the

composite film to dynamic loading with pressures in the 0.07–100 kPa range. (C) Variation in the sensor output voltage with pressure. (D) Response of composite film (top) and P(VDF-TrFE) with Au electrode (bottom) to periodic heating and cooling at 10 K... ” (page 10)

Comment 6: For discussion, very thick electrodes (>100 um) are required to achieve a high d_{33} as shown in Fig 2H. Compared to normal metal electrode with a thickness in the range of several hundred nanometers to several micrometers, It is very challenging to pattern the polymer electrodes using traditional wet or dry etching methods.

Response: Thanks for the reviewer’s thought-provoking comment. To address this comment, we were trying to reduce the thickness of PEDOT:PSS electrode by various methods, including using 38 vol% DMSO solution ($V_{\text{PEDOT:PSS}}/V_{\text{DMSO}}=38\%$), changing the drying temperature and time. It has been shown that a high d_{33} can be achieved with a relatively thin PEDOT:PSS electrode. It shows that it is possible to reduce the thickness of the PEDOT:PSS electrode to obtain a larger piezoelectric coefficient.

Fig. R2 The d_{33} as a function of the thickness of the composite film using a DMOS solution of $V_{\text{PEDOT:PSS}}/V_{\text{DMSO}}=38$ vol% and 5 vol%

Minor comments

Comment 7: In the sentence of “The conventional poly(vinylidene fluoride-co-trifluoroethylene) copolymer (P(VDR-TrFE)) has a low sensor voltage output because of its relatively low piezoelectric and pyroelectric coefficients”. “P(VDR-TrFE)” -> “P(VDF-TrFE)”

Response: Thanks for the reviewer’s correction. We have changed the “P(VDR-TrFE)” to “P(VDF-TrFE)” in the revised manuscript as follows.

“The conventional poly(vinylidene fluoride-co-trifluoroethylene) copolymer (P(VDF-TrFE)) has a low sensor voltage output.....” (page 2)

In addition, the format of the units in the revised figure has been adapted to the format used in the article.

Responses to Reviewer #3 (Remarks to the Author):

Paper on influencing a network interconnection interface to enhance properties. The novelty stems from the network/conductivity electrode.

We appreciate the reviewer's acknowledgement of our work. All questions and concerns were addressed point-by-point.

Comment 1: "For a fixed polymer thickness, expanding the contact area between the polymer and electrodes causes the capacitance to increase" - does the effective thickness also change/decrease due to the conductive electrode?

Response: Thanks for the reviewer's comments. We believe that the effective thickness of has not been changed for the following reason.

1) The cross-sectional image shows that the thickness of the P(VDF-TrFE) did not change with or without the PEDOT:PSS electrode (see below).

Fig. R1 The cross-section of the P(VDF-TrFE)/PEDOT:PSS composite film

2) The increase in dielectric coefficient comes from the heterogeneous conductivity of the network.

“According to the Koop’s model⁴⁸, the dielectric heterogeneous network conductivity can be considered as a conducting PEDOT:PSS separated by the high-resistance P(VDF-TrFE). The space charge created by the stress due to the piezoelectric effect builds up at the P(VDF-TrFE), controlling the available free charge carriers at the P(VDF-TrFE) and leading to a higher dielectric constant.” (page 6)

Comment 2: Was a poling process applied to the device. This is not clear in the paper, and if not, how is the polarisation achieved? I also see no classical polarisation-electric field hysteresis loops? This would be desirable.

Response: The P(VDF-TrFE) films were poled before they were applied to the devices. To address this comment, the poled condition was added to the revised manuscript as follows.

“To fabricate the P(VDF-TrFE)/PEDOT:PSS composite film, the P(VDF-TrFE) film with the thickness of 80 μm was first poled by corona poling at room temperature.” (page 12)

The polarization-electric field (P - E) hysteresis loops of the P(VDF-TrFE) films with Au, PEDOT:PSS electrode were added in the supporting information (Fig S3) as follows. The P - E loops of the P(VDF-TrFE)/PEDOT:PSS composite film with $d_{33}>60$ pC N^{-1} were also measured. However, the leakage current was large and was not included in the revised manuscript.

“The polarization-electric field hysteresis loops of the P(VDF-TrFE) films with Au ($d_{33}=20$), PEDOT:PSS electrode proved that the composite film has ferroelectric properties (Fig. S3). The leakage current of the composite film increases with the increase of d_{33} , and the reason will be explained later.” (page 4)

“At a high DMSO volume fraction, the NII may contact and lead to neutralization of the positive and negative charges (Fig. S8A), eventually leading to a decrease in d_{33} and an increase in leakage current.” (page 7)

Fig. S3 Polarization-electric field hysteresis loops of the P(VDF-TrFE) film with Au electrode ($d_{33}=20$) and PEDOT:PSS electrode ($d_{33}=43, 50, 57$).

Comment 3: It is stated the d_{33} is measured “using a quasistatic d_{33} instrument (ZJ-3AN) - in the discussion on Figure 2 it is stated “the calculation, the piezoelectric coefficient is determined by the permittivity.” - this could be more clear. It would be good if the d_{33} could be clearly described as the value is large >80 pC/N

Response: Thanks for the reviewer’s comments. According to these comments and suggestions, the calculation details of the d_{33} of the composite film were given in the revised manuscript. In addition, the calculated d_{33} whose value is greater than > 80 pC/N were added to Fig. 2 as follows.

“Fig. 2ACalculated and experimentally measured (A) piezoelectric coefficient and (B) pyroelectric coefficient of composite films at different NII thickness ratios.” (page 8)”

“The piezoelectric properties of the film can be obtained by

$$d_{33} = 2\varepsilon_{33}\varepsilon_0 \left[Q_{11} - \frac{2s_{11}Q_{12}}{s_{11} + s_{12}} \right] P, \text{ where } Q_{11} \text{ and } Q_{12} \text{ are the electrostrictive coefficients,}$$

s_{11} and s_{12} are the elastic compliances.” (page 15)

Comment 4: Is there a danger of developing a short-circuit in the material?

Response: There is no danger of short-circuit in the materials. In the short-circuit model, the short-circuit current of the P(VDF-TrFE)/PEDOT:PSS film under the stimuli of the stress and temperature was measured and added to the revised supporting information (Fig. S10) as follows. It can be seen that the short-circuit current is low due to the high resistance of the P(VDF-TrFE) film.

Fig. S10 (A) Piezoelectric and (B) pyroelectric current of the composite film as a function of the pressure and temperature, respectively.

Comment 5: “The cross-sectional area of the applied force determines the $|d_{33}|$ of P(VDF-TrFE).” - this is an unusual comment as most d_{33} meters apply a small area (almost point force) and collect charge - and apply force to the same surface as the charge is collected removes the effect of area. This links to my comment on the lack of clarity on how d_{33} was measured.

Response: Thanks for the reviewer’s comments. We agree with the reviewer that most d_{33} meters apply a tiny area, collect charge, and then apply a force to the same surface, eliminating the area effect. In this work, the piezoelectric coefficient of polarization P(VDF-TrFE) film was measured using a quasi-static d_{33} instrument (ZJ-3AN, IACAS) at room temperature. The measuring head of the ZJ-3AN contains an electromagnetic force drive that generates a low-frequency (110 Hz) alternating force (0.5 N). The contact of NII obtained from the experiments (Fig. S8A) was applied to explain the reason for the decrease of d_{33} at high DMSO volume fraction. In addition, we calculated the dimensionless surface potential in the composite film as a function of the NII thickness ratio and found that neutralization of the positive and negative charges could be the reason for the decrease of d_{33} at high DMSO volume fraction (Fig. S8B). To address this comment, the following change was added in the revised manuscript: “At a high DMSO volume fraction, the NII may contact and lead to neutralization of the positive and negative charges (Fig. S8A), eventually leading to a decrease in d_{33} and an increase in leakage current. This was confirmed by the calculated dimensionless surface

potential obtained from Maxwell's equation (Fig. S8B), which decreases when the NII thickness ratio increases from 81.3% to 87.5%.” (page 7)

“The piezoelectric coefficient of the polarization P(VDF-TrFE) film after was measured using a quasistatic d_{33} instrument (ZJ-3AN, IACAS) at room temperature. The measuring head of ZJ-3AN contains an electromagnetic force drive that generates a low-frequency (110 Hz) alternating force (0.5 N).” (page 13)

Comment 6: The application of the material at the end is interesting but not vital and innovative - the key is to demonstrate enhancement in materials properties.

Response: Thanks for the reviewer's good suggestion. Fig. 4 was shortened according to the reviewer's suggestion. To prove that the composite film can be used as an ultra-sensitive mechanical/thermal sensor for health monitoring, an application for monitoring finger pulse, which is difficult to detect with a piezoelectric sensor, was added to the revised Fig. 4 (see below).

“**Fig. 4 Application of the P(VDF-TrFE)/PEDOT:PSS composite film as an ultrasensitive sensor.** A Pulse waveform from sensor monitoring of a thumb (top) and subject's wrist (bottom). B Magnified view of one cycle of the electrical signal. C Speech recognition based on monitoring laryngeal activity while speaking different words.” (page 11)

“Next, the fabricated real-time physiological monitoring device was placed on an adult human to demonstrate the potential and usability of the composite film in health monitoring. The device was attached to the thumbs up or wrist of various volunteers to detect subtle fluctuations in arterial blood pressure (Fig. 4A). Frequency and amplitude of pulses whichi obtained form are accurately reproduced in terms of distance between adjacent peaks and average peak amplitude in real time. Normal heart rate remains at

72 and 79 beats per minute (bpm) for different volunteers. The measured values are close to the performance of a commercially available device (inset in Fig. 4A), demonstrating the ability of the prepared device based on a composite film for real-time physiological monitoring. The detailed waveforms in Fig. 4B show characteristic peaks corresponding to percussion waves (p-waves), tidal waves (t-waves), and dicrotic waves (d-waves) in the human pulse, consistent with other work^{61,62}. The prepared sensor was attached to the participant's throat to test the possibility of devices as a word recognition system. Vibration of the throat was detected when different words were spoken. A single peak of output voltage was present when the monosyllabic word "Hi" was spoken, whereas the multisyllabic word "Hello" elicited a multimodal shape (Fig. 4C). Finally, the sensor was used to monitor finger flexion (Fig. S13A) and hand grasping (Fig. S13B and S13C). In addition, a sensor array was prepared to prove that the sensor array can map the dispersion of tactile sensations (Fig. S14, S15)." (page 10-11)

REVIEWER COMMENTS

Reviewer #1 (Remarks to the Author):

The authors have answered most of my questions. No more comments.

Reviewer #2 (Remarks to the Author):

Though the paper has been improved during the 1st turn revision, there are still several confusing points to be clarify.

1. The author mentioned that “The piezoelectric response time of the composite film (0.06 S) is shorter than that of the mental electrode (0.1S), indicating that the composite film has better response properties.” How did author define “response time”? Please clearly show the characterization procedure for response time. Could the author give a theoretical analysis that the device with PEDOT:PSS electrode exhibits quicker response than the one with metal electrodes?

2. How did the author do the poling process for the device with metal electrodes? Is it also a corona discharging process? Why don't the author do the poling process after the formation of electrodes?

3. The author mentioned that “This piezoelectric current of NII-derived penetrating electrode (~76 nA) is larger than the previously reported penetrating electrodes (~20 nA) at 5 kPa.” Please keep in mind that the piezoelectric current is proportional to the effective surface area of the sensor. The device in this manuscript has a size of 10mm*10mm. However, the device reported in reference is only 0.64mm². Please make a rational comparison with the previously reported devices.

4. I suggested that “Figure 3 show the result in high pressure range (larger than 1kPa). I strongly recommend including supplementary results that show the sensing performance in low pressure ranges such as 0~100Pa.” However, the author just add the single point of 0.07kPa. Please clearly show the result in low pressure region. Is it linear with the pressure?

6. As the device is encapsulated with PDMS, the author should give the measurement result to show the effect of PDMS to sensor performance. Is the piezoelectric output same to the device without PDMS encapsulation?

Reviewer #3 (Remarks to the Author):

Generally good reply to comments, the polarisation-electric field loops are not fully saturated but as described in the text there is some conductivity.

In the reply, new text has been added.

"The polarization-electric field hysteresis loops of the P(VDF-TrFE) films with Au ($d_{33}=20$)..." - the d_{33} needs to have units.

List of Responses to the Reviewers' Comments

We would like to thank the reviewers for their valuable comments and suggestions on this manuscript, which greatly improves the quality of our manuscript. Following these comments and suggestions, we have made careful revisions to our previous manuscript (marked in red colour), and provide response to the comments (marked in blue colour) point-by-point as follows:

Responses to Reviewer #1 (Remarks to the Author):

The authors have answered most of my questions. No more comments.

We appreciate the reviewer's acknowledgement of our manuscript.

Responses to Reviewer #2 (Remarks to the Author):

Though the paper has been improved during the 1st turn revision, there are still several confusing points to be clarify.

We appreciate the reviewer's recognition of our work. All questions and concerns were addressed point-by-point.

Comment 1: The author mentioned that "The piezoelectric response time of the composite film (0.06 S) is shorter than that of the metal electrode (0.1S), indicating that the composite film has better response properties." How did author define "response time"? Please clearly show the characterization procedure for response time. Could the author give a theoretical analysis that the device with PEDOT:PSS electrode exhibits quicker response than the one with metal electrodes?

Response: We apologize the misunderstanding induced from our claim on the piezoelectric response time. We re-term it as output response time of sensor device rather the response simply from piezoelectricity. More specifically, the composite film containing PEDOT:PSS electrodes and P(VDF-TrFE) piezoelectric materials can be considered as a sensor. The response time of the devices is determined by the rise time of the output voltage of the sensor, which can be deduced from the shape of the output voltage. In particular, the response time of the sensor is defined as the time required for the response voltage to rise from 10% to 90%. We apologize that our previous

statements may have caused possible misunderstandings among readers. To response this comment, related discussions and two new figures (Fig. S10 and S11) were added in the revised manuscript and supplementary information (see below):

“The output signal of the P(VDF-TrFE)/PEDOT:PSS film as a sensor was studied with a total thickness of 230 μm and $|d_{33}|=86 \text{ pC N}^{-1}$, and the solvent of PEDOT:PSS was DMSO..... The rise time of the composite film, defined as the time required for the response voltage to rise from 10% to 90%⁵⁴(0.06 S) (Fig. S10A and S10B), is shorter than that of the metal electrode (0.1 S), which is partly due to the higher rate of domain evolution in the composite film (Fig. S11, see supplementary information for details). This indicates that the composite film has a better piezoelectric response.” (page 8-9)

“The response time of output signal of the composite film induced by thermal stimulus (1.77 S) is shorter than that of the Au electrode (3.80 S) (Fig. S10C and S10D) because the PEDOT:PSS electrode can absorb more radiant heat energy and increase the dT/dt .” (page 9)

“Fig. S10 Response time of output signal of P(VDF-TrFE) film induced by mechanical stimulus with (A) PEDOT:PSS, (B) Au electrode. Response time of output signal induced by thermal stimulus with (C) PEDOT:PSS, (D) Au electrode.”

In addition, we provide a theoretical analysis of response time of piezoelectric sensor from domain movement. Here the piezoelectric sensor consists of an electrode, piezoelectric material and an interface between them. We simplified the case by only considering these factors which can affect the response time of the piezoelectric sensor. The influence of other factors on the response time, such as the electrode materials and

microstructures is far beyond the scope of our manuscript which should be studied in future studies.

“Theoretical analysis of response time of piezoelectric sensor from domain movement

To study the influence of the NII on the response time of piezoelectric signal, the phase field simulation is applied to simulate the movement of the domain under pressure. In the phase field simulation, the total number of time steps represents the development rate of the domain structure of P(VDF-TrFE) under the external field. The larger value of the time step, the slower the evolution rate will be.

The simulated and actual domain evolution time can be determined by the kinetic coefficient (L) in the TDGL equation, which can be obtained for domain wall dynamics experiment. Since the value of L is not determined, the normalized time and polarization is used in Fig. S11. The simulation results show that the evolution time of the P(VDF-TrFE) film with NII under external forces (133 steps) is shorter than that of the film without NII (276 steps), which is in line with our experimental results. Therefore, the response time of the P(VDF-TrFE) with PEDOT:PSS electrode is lower than that of the metal electrode, which can be explained to some extent from the perspective of domain movement.

Actually, a piezoelectric sensor consists of an electrode, piezoelectric materials and an interface between them. This means that these factors can affect the response time of the piezoelectric sensor. This theoretical analysis of the response time of the piezoelectric sensor only considers the movement of the domain under mechanical load⁴, the influence of other factors, such as the electrode materials and microstructures, should be further investigated.”

Fig. S11 Domain revolution of P(VDF-TrFE) film with and without NII as a function of normalized time obtained from phase-field simulation.

Comment 2: How did the author do the poling process for the device with metal electrodes? Is it also a corona discharging process? Why don't the author do the poling process after the formation of electrodes?

Response: Thanks for the reviewer's comments. The P(VDF-TrFE) film was first poled by corona poling at room temperature before PEDOT:PSS and metal were applied to it. There is a difference in the piezoelectric coefficient of the poled P(VDF-TrFE) ($18\sim 28$ pC N⁻¹) due to the uncontrollable changes in the experimental parameters, such as temperature variations of the oven. To ensure that the P(VDF-TrFE) film used has the same d_{33} , the P(VDF-TrFE) was poled before the electrode was fabricated. To address this comment, the following description about the poling process was added to the revised manuscript.

“To ensure that the P(VDF-TrFE) film used show the same d_{33} values (~ 20 pC N⁻¹), the P(VDF-TrFE) film with a thickness of 80 μm was poled by corona poling at room temperature before fabricating the electrode.” (page 13)

“For metal electrode devices, the poled P(VDF-TrFE) film was sputter-coated using a small ion sputterer (DM200, Hefei Dingshuo).” (page 13)

Comment 3: The author mentioned that “This piezoelectric current of NII-derived penetrating electrode (~ 76 nA) is larger than the previously reported penetrating electrodes (~ 20 nA) at 5 kPa.” Please keep in mind that the piezoelectric current is proportional to the effective surface area of the sensor. The device in this manuscript has a size of 10mm*10mm. However, the device reported in reference is only 0.64 mm². Please make a rational comparison with the previously reported devices.

Response: Thanks for the reviewer's comments. To response this comment, a new figure about the comparison of piezoelectric and pyroelectric current density between this work and previously reported devices was added to the revised supplementary information as follows. In addition, the following discussion was also added to the revised manuscript.

“Note that the piezoelectric/pyroelectric current depends in part on the effective surface area of the sensor. The comparison of piezoelectric and pyroelectric current density (Fig.

S14) indicates that the composite film has high piezoelectric and pyroelectric output properties.” (page 9)

Figure S14 Comparison of (A) piezoelectric and (B) pyroelectric current density between this work and previously reported devices⁵⁻¹².

Comment 4: I suggested that “Figure 3 show the result in high pressure range (larger than 1kPa). I strongly recommend including supplementary results that show the sensing performance in low pressure ranges such as 0~100 Pa.” However, the author just add the single point of 0.07 kPa. Please clearly show the result in low pressure region. Is it linear with the pressure?

Response: Thanks for the reviews comment. To response this comment, the piezoelectric current and voltage of the composite film under the pressure of 25, 50 and 75 Pa was measured, respectively, which is shown as Fig. S13 in our revised supplementary information. Piezoelectric voltage of the composite film under the pressure of 50 Pa was added to Fig. 3B.

“In Fig. 3C, the output voltage of the composite film is plotted as a function of applied pressure, and the piezoelectric response at low pressure of 25-100 Pa is shown in Fig. S13. These results confirm that the composite film exhibits a linear relationship between these two variables” (page 9)

“Fig. 3 Piezoelectric and pyroelectric voltage output of PEDOT:PSS/P(VDF-TrFE) with the NIL.(B) Response of the composite film to dynamic loading with pressures in the range of 0.025–100 kPa.”

“Figure S13 Piezoelectric (A) voltage and (B) current of the composite film as a function of the pressure of 25, 50, 75 and 100 Pa.”

Comment 6: As the device is encapsulated with PDMS, the author should give the measurement result to show the effect of PDMS to sensor performance. Is the piezoelectric output same to the device without PDMS encapsulation?

Responses: To address this comment, the temperature and pressure sensor performance of the devices with and without PDMS encapsulation was measured which is shown in Fig. S16. In addition, the discussion of the effect of PDMS on sensor performance was added to the revised supplementary information as follows.

“The pressure and temperature sensor performance of the devices with and without encapsulation shows that PDMS slightly reduces the output voltages for both pyroelectric and piezoelectric devices (Fig. S16).” (page 10)

“Influence of flexible substrate on the sensor property

When external forces are applied to the encapsulated device, the flexible PDMS with their low thermal conductivity reduce the actual force and temperature applied to the device, resulting in slightly lower output voltages for both pyroelectric and piezoelectric devices (Fig. S16).”

Figure S16 (A) Pressure and (B) temperature sensor performance of the devices with and without PDMS encapsulation.

Responses to Reviewer #3 (Remarks to the Author):

Generally good reply to comments, the polarisation-electric field loops are not fully saturated but as described in the text there is some conductivity.

We appreciate the reviewer’s acknowledgement of our manuscript. The comment was addressed point-by-point.

Comment : In the reply, new text has been added.

"The polarization-electric field hysteresis loops of the P(VDF-TrFE) films with Au ($d_{33}=20$)..." - the d_{33} needs to have units.

Responses: Thanks for the reviewer’s comment. To response this comment, the units of d_{33} have been inserted into the revised manuscript as follows.

“The polarization-electric field hysteresis loops of the P(VDF-TrFE) films with Au ($|d_{33}|=20$ pC N⁻¹), PEDOT:PSS electrode proved that the composite film has ferroelectric properties (Fig. S3).” (page 4)

REVIEWERS' COMMENTS

Reviewer #2 (Remarks to the Author):

The authors have answered all my questions. No more comments.

List of Responses to the Reviewers' Comments

We would like to thank the reviewers for the comments on this manuscript. Following these comments and suggestions, we have provide response to the comments (marked in blue colour) point-by-point as follows:

Responses to Reviewer #2 (Remarks to the Author):

Comment: The authors have answered all my questions. No more comments.

Response: We appreciate the reviewer's acknowledgement of our manuscript.